# Reasoning With Hierarchical Symbols: Reclaiming Symbolic Policies For Visual Reinforcement Learning

## Abstract

Deep vision models are nowadays widely integrated into visual reinforcement learning (RL) to parameterize the policy networks. However, the learned policies are overparameterized black boxes that lack interpretability, and are usually brittle under input distribution shifts. This work revisits this end-to-end learning pipeline, and proposes an alternative stage-wise approach that features hierarchical reasoning. Specifically, our approach progressively converts a policy network into the interpretable symbolic policy, composed from geometric and numerical symbols and operators. A policy regression algorithm called *RoundTourMix* is proposed to distill the symbolic rules as teacher-student. The symbolic policy can be treated as discrete and abstracted representations of the policy network, but are found to be more interpretable, robust and transferable. The proposed symbolic distillation approach is experimentally demonstrated to maintain the performance and "denoise" the CNN policy: on six specific environments, our distilled symbolic policy achieved compelling or even higher scores than the CNN based RL agents. Our codes will be fully released upon acceptance.

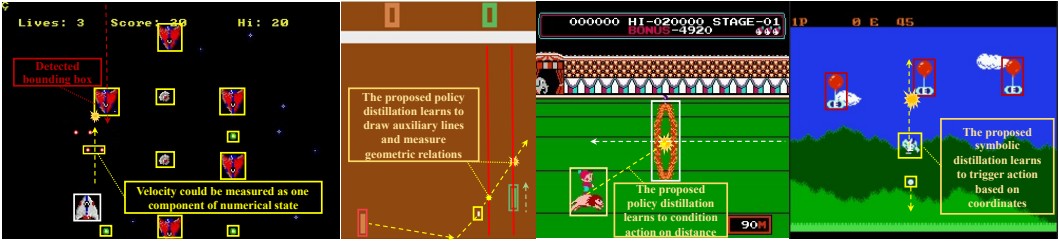

Figure 1: Four example environments adopted for distilling the CNN policy network knowledge into the symbolic policy. From left to right are: Airstriker-Genesis, Pong, CircusCharlie, Seaquest.

## 1 Introduction

Reinforcement learning (RL) is effective to explore complex environments through interactions (Kaiser et al., 2019). A central target in building stronger RL algorithms is to learn reliable and generalizable policies, robust to unseen real-world data with possible shifts. This is a difficult task, especially when the input space is high-dimensional or continuous, e.g., the visual data, and the sampled observations during training are hence destined to be extremely sparse. To learn sophisticated policies from those data, it is common to parameterize policies as *numerical* predictors, e.g, neural networks (Francois-Lavet et al., 2018). Despite their success, the neural network based RL algorithms require prohibitive amounts of data to train, and the resultant rules are "black boxes" that are hardly interpretable (Mnih et al., 2013; Heuillet et al., 2021) nor stable under input domain shifts (Schmeckpeper et al., 2020).

To mitigate this gap, let us take inspirations from ourselves. Humans integrate information at multiple levels of abstraction: from raw sensory, to concept symbols, and to more underlying

"computational" mechanisms (Timpf et al., 1992). The hierarchical and progressive integration of concept abstraction serves as the key for intermediate states to enable interpretable logical reasoning as well as for generalization: the latter because appropriate abstraction can eliminate nuisances (Krawczyk et al., 2011). Therefore, we are motivated to program our visual RL policies into an alternative representation of *symbolic* rules. Compared to its numerically parameterized counterparts, a symbolic rule is significantly lighter-weight, likely more interpretable, and potentially more generalizable.

Abstracting pure *symbolic* rules from data is not new. The family of symbolic regression (SR) methods (Cranmer et al., 2020; Runarsson & Jonsson, 2000; Gustafson et al., 2005; Orchard & Wang, 2016) have been studied to directly search from the discrete symbolic space to compose target equations. Different from conventional regression techniques that optimize a pre-specified model structure, SR infers both model structures and parameters from data. Most popular SR algorithms rely on genetic programming to evolve symbolic math equations from scratch (Runarsson & Jonsson, 2000; Gustafson et al., 2005; Orchard & Wang, 2016). However, compared to the differentiable learning of numerical policy networks, those SR methods are slow on large/complicated problems and rely on many heuristics to work (Runarsson & Jonsson, 2000; Gustafson et al., 2005; Orchard & Wang, 2016). Hence, naively plugging those methods as policy learners for visual RL, e.g., to initialize a population of simple symbolic policies and then gradually mutate/select better candidates while interacting with the environment, will be computationally prohibitive (Runarsson & Jonsson, 2000; Gustafson et al., 2005; Orchard & Wang, 2016), in addition to suffering from the sparse rewards.

We thus propose to synergize the numerical and symbolic policy learning, hoping to gain the best from both worlds. Inspired by the idea of "knowledge distillation" (Hinton et al., 2015), we propose a symbolic policy distillation approach: it does not directly compose and evolve the symbolic policy through RL interaction, but instead, distills a symbolic policy from the behavior of a numerical policy network as the "teacher". After the teacher is learned via conventional RL, our approach extracts a symbolic form that approximates the relationship between the input visual observations and the output actions of the teacher. A key barrier here is, however, the fact that the teacher's input is an image, while a symbolic policy cannot easily consume a high-dimension continuous input. A robust parsing mechanism is thus needed to abstract an image into a set of symbolic input operands, on top of which more symbolic operators can be defined to compose them. That requires us to cross the famous "semantic gap" (Hein, 2010) in vision, by properly discretizing the perception-decision procedure.

In view of the above challenges, we study the progressive discrete symbolization of representations, from the continuous visual observations to the discrete actions, and accordingly design a symbolic policy distillation procedure that leverages the resultant hierarchical symbols. The progressive symbolization for visual RL bridges the raw image input, the desired symbolic input operands, and the output action space. To achieve so, we first use an object detection module to achieve the "first level" of discretization, i.e., from the input image to a discrete set of objects (both classes and locations). We then proceed to the "second level", to (iteratively) search the symbolic rule that maps the input set of objects to the output set of actions. The second level is accomplished using a symbolic teacher-student distillation algorithm named *RoundTourMix*. Our entire process yields a hierarchical, explainable, and generalizable symbolic rule in the form of symbolic compositions, that can be treated as discrete and abstracted representations of the visual policy network.

As observed through extensive experiments, our symbolic policy distillation approach captures the underlying structure in a white-box form through data-driven experience, and meanwhile gets rid of visual domain nuisances through state abstraction (e.g., detection). In particular, the distilled symbolic policy leverages the representation of object location, velocity and class information, which naturally group two scenarios (visual scenes) together if the same logical specifications are congruent even if the scenes' pixel-level attributes are markedly different. The distillation procedure also requires orders of magnitude less data than policy neural network training. Our approach paves a new way to make the visual RL more interpretable, reliable and generalizable, and also narrows the gap between numerical and symbolic approaches in RL. Our concrete technical contributions are summarized as follows:

- We propose a novel progressive discretization of the perception-decision procedure with three logical hierarchies: from a raw image to its object-level abstraction (called "geometric symbols" hereinafter), to the numerical state and discrete action spaces.

- We propose a novel policy regression algorithm that distills the teacher behaviors of policy networks into symbolic rules. Our approach termed *RoundTourMix* is gradient-free and data-efficient to train, and the output rule enjoys good interpretability and reliability.

- Through performance/interpretability/transferability evaluation, our distilled symbolic policies demonstrate comparable or even stronger effectiveness in seven highly challenging visual RL environments. They achieve the comparative or higher scores with interpretable actions while generalizing better on the new environments than their teacher policy networks.

## 2 RELATED WORKS

Our work tightly intersects with interpretable reinforcement learning, and mathematical symbolic regression. In reinforcement learning, the way to boost interpretability can be roughly categorized into **(a)** intrinsic interpretability based methods and **(b)** post-hoc explanation based methods. The **Intrinsic interpretability** based methods require the learned models to be self-understandable by learning over interpretable architectures. Among the existing methods, the authors in Lyu et al. (2019) proposed to learn an interpretable policy online, Zambaldi et al. (2018); Jiang & Luo (2019); Dong et al. (2019) to introduce architectural inductive biases to learn a interpretable policy, Dittadi et al. (2020) to perform width-based planning on relevant features extracted from game frames, and Ma et al. (2021) preserves interpretability by learning a set of logical rules. Other methods also leverage imitation learning (Verma et al., 2018; 2019; Bastani et al., 2018). For **post-hoc explanation** based methods, the papers in Zahavy et al. (2016); Greydanus et al. (2018); Gupta et al. (2019) propose to use t-SNE and saliency map to explain the learned policy. (Liang et al., 2015) proposes to capture key features of a DQN into practical linear representations. Other methods include attention based method (Shi et al., 2020), visual summary based method (Sequeira & Gervasio, 2020), reward decomposition based method (Juozapaitis et al., 2019), casual model based method (Madumal et al., 2020), Markov chain based method (Topin & Veloso, 2019) and case-based expert-behaviour retrieval method (Ontanón et al., 2007).

Our symbolic distillation falls within the post-hoc methods, as we first learn a CNN policy that enjoys the neural network's free form optimization, then distills it into an explainable symbolic policy. Connecting to existing methods that follow this path, the work in (Coppens et al., 2019) distill the learned policy into a soft decision tree. Their visual observations are coarsely quantized into $10 \times 10$ cells, and the soft decision tree policy is learned over the 100 dimensional space. Garnelo et al. (2016); Garcez et al. (2018) also learn symbolic RL algorithms, but the states of the environments they considered are clean and simple, and are not as challenging as video games.

Symbolic Regression (SR) (Runarsson & Jonsson, 2000; Orchard & Wang, 2016; Schmidt & Lipson, 2009; Cranmer et al., 2020; Cranmer, 2020; Petersen et al., 2019; Gustafson et al., 2005) is a recently emerging and promising approach for discovering underlying symbolic rules of the observed data. In discovering the mathematical equations, several algorithms adopted the gradient-free genetic programming method. These algorithms generate populations of candidates equations, mutate them and select better candidates, then repeat this evolution procedure until certain thresholds of maximum interation number or performance level are met. Examples include Eureqa (Schmidt & Lipson, 2009) and PySR (Cranmer et al., 2020; Cranmer, 2020), and a few others (Runarsson & Jonsson, 2000; Orchard & Wang, 2016; Gustafson et al., 2005). Some other methods leverage the deep neural network to parse out equation trees, such as (Petersen et al., 2019; Bello et al., 2017).

## 3 THREE TYPES OF IMAGE DATA REPRESENTATIONS

In this work, we focus on the visual Reinforcement Learning (RL) domain, and distill the teacher agent behavior into a symbolic form. We use the *Gym Retro games* (Nichol et al.) as primary examples to study policy distillation. For the most common visual RL case, the agent is modeled by an end-to-end neural network (usually based on a convolutional neural network, or CNN) that maps the image type data to the action. As pixel level data representation, the raw image offers rich low level information but without abstraction. On the other hand, as a high level decision regime, the symbolic rule takes robust abstracted symbols as its input operands.

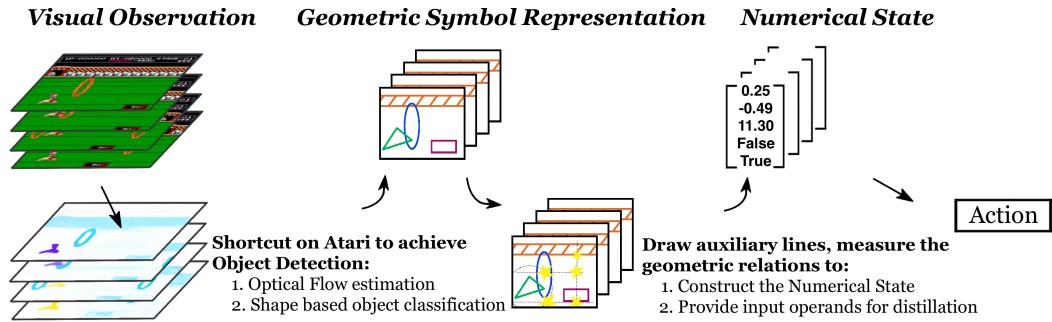

Figure 2: The discretized perception-decision procedure, with three representation types (see Sec. 3).

To bridge this gap, it is hence necessary to summarize the image pixels into symbolic operands (also refered to as geometric objects in this work) as required by the symbolic policy. Besides the object level abstraction of the raw image, richer numerical form representations are also needed, as the numbers are the basic building blocks to effectively represent knowledge and conduct learning. Summarizing these motivations, we introduce three types of representations, the **visual observation**, the **geometric symbol representation**, and the **numerical state**.

The initially obtainable representation is the **visual observation**, which contains the raw information and serves as the input for the CNN based model. With these visual observations, we run a (pre-trained and fixed) object detection algorithm to obtain object lists in the images. The object detection module will provide us the class, location, as well as the velocity of every single detected object in the image sequence. The output of the object detection module is the second type of image data representations, which we term as the **geometric symbol representation**. The geometric symbol representation is the desired symbol abstraction of the pixel level raw image, and supplies the input operands for the symbolic distillation to work on. We use optical flow to segment and estimate the velocity of the objects, and further classify the objects based on their shapes. More details are in Appendix B. Eventually, the third type of representation is the **numerical state**, which is obtained by drawing auxiliary lines, and measuring/quantizing certain relationships of the geometric symbols (distances, orientations, etc). The numerical state is then used to predict the final action.

Overall as illustrated in Fig. 2, in order to parse a raw image into a symbolic policy, the perception-decision procedure is discretized into the following steps: `visual observation →` `geometric symbol representation → numerical state → action`.

## 4 LEARNING TO DISTILL TEACHER BEHAVIOR INTO SYMBOLIC POLICY

With the discretized decision procedure defined above, the remaining question is how to concretely distill the teacher behavior into the symbolic policy. In what follows, we summarize the learning/distillation objectives in section 4.1, then describe the components of symbolic search space in section 4.2, and finally describe the distillation algorithm in section 4.3.

### 4.1 THE LEARNING OBJECTIVES

Over the three representations discussed above, a hierarchical symbolic reasoning procedure can be built. In this work, we posit that the step `[visual observation →` `geometric symbol representation]` could be robustly implemented and is reliable for the usage of future steps, therefore, we set it as learning-less. Based on the observed geometric symbol representation, we learn the rest of reasoning steps `[geometric symbol representation` `→ numerical state → action]` through the proposed symbolic distillation.

Intuitively, there are three reasoning components subject to learning: ❶ **what geometric relationship to measure** out of the geometric symbol combinations, so as to compose the numerical state; ❷ **what is the condition to take action** based on the measured numerical state; and ❸ **how to take action**, under the current condition. In this research, the learning targets ❷ and ❸ are usually found to be stacked with each other in a hierarchical way, yielding a tree-shaped decision procedure (Sutton et al., 1999; Dietterich, 2000; Botvinick et al., 2009). To connect these learning

targets with the discretized decision procedures in section 3, ❶ determines the learning target for the [geometric symbols → numerical state] step, and ❷ and ❸ determine the learning target for the [numerical state → action] step. In the following section, we define and construct the basic building block to compose these learning targets.

## 4.2 SYMBOLIC POLICY SEARCH SPACE

To make symbolic policy distillation executable, we have to answer two open questions: what are the available building blocks to construct symbolic policy, and how can we construct it? In the following, we answer them by defining the symbolic search space, which contains the geometric operator search space and the numerical operator search space.

**The geometric operator search space**. The geometric operator search space is designed for the learning target ❶ in section 4.1, and the components within this space are geometric operator combinations applied to detected objects. To enable stronger expressiveness, we include two types of geometric operators: the *auxiliary line drawer* and the *geometric attribute measurer*. The *auxiliary line drawer* takes the input of detected objects, and draw auxiliary lines according to certain given descriptions. This type of operators include `velocity_extension` (extend the line along certain object's velocity direction, and keep track of it), `static_line_drawer` (drawing static vertical/horizontal/tilt auxiliary lines), and `intersection_marker` (mark the intersection between the auxiliary lines and/or detected objects, and keep track of it). The newly drawn auxiliary lines/intersection points are added into the collection of the geometric symbol representation.

For the second type of geometric operators, the *geometric attribute measurer* is responsible for evaluating the exact geometric relationships given specified geometric objects. The evaluated geometric relationships could be real/boolean values, or another geometric object. The real valued operators include `velocity_of` (certain given object), `location_of` (certain given object), `distance_between` (two given objects), `orientation_between` (two objects against each other). The boolean valued operators include `has_intersection` (detect intersection between two objects or an object and an auxiliary line), `is_moving`, `is_changing_shape` (inspect one given object). Another special operator is `find_nearest_object`, which returns the nearest class-$i$ ($i$ is given) object from a reference object in the current observation, and the attributes of this object could be further measured by other geometric attribute measurer. This operator is found to be frequently used, as it is usually the case that when multiple objects are present, the agent tends to deal with the nearest object first.

**The numerical operator search space**. In correspondence with the geometric search space, the numerical search space is designed for learning targets ❷ and ❸, and its components are numerical operator combinations applied to the variables in the measured numerical state. There are also two types of numerical operators. The first type is the bool function/logical operators, which include $is(x < y), is(x \leq y), is(x = y), a \mid b, a \& b, \neg a$, and conditional combinations: $a \star x + (\neg a) \star y$. The second type is the math operators, which include $+, -, \star, /, \sin, \cos, \tan, \cot, (\cdot)^2, (\cdot)^3, \sqrt{\cdot}, \exp, \log$.

To illustrate with two concrete examples, in the environment *CircusCharlie* (illustrated in Fig. 1 and Fig. 2), the agent's target is to jump right into the fire ring and over the pot. One sample distilled symboblic policy is presented in Fig 3. The equation 1 also represents a subtree of Fig 3 for simplicity. In Eq 1, $x_{\text{ring}}, x_{\text{pot}}$ and $x_0$ are the $x$-coordinate of the detected fire ring, pot and the protagonist objects. The distilled policy allows the agent to only jump when the condition $\neg(10 < x_{pot} - x_{ring} < 50) \times (x_{ring} - x_0 < 40)$ is met. In *Pong* (illustrated in Fig. 1), the agent needs to control the right racket so as to hit the pong, which can collide and bounce with the walls. One sample distilled symbolic policy is in Eq. 2, where $x_{\text{pong}}$ is the $x$ coordinate of pong, S is the intersection of the speed orientation and the right edge (drawn and tracked by the *auxiliary line drawer*). Such policy allows the agent to ignore the bouncing procedure of the pong, only pay attention to the speed orientation of the pong when it is near the edge ($x_{\text{pong}} > 40$). More examples of distilled policies and their interpretations are in Appendix. C.

One possible difficulty of the symbolic distillation procedure is the huge size of the search space: it grows exponentially with longer time span. Denote the number of symbolic compositions in one frame as $N$. If one consider stacking $T$ consecutive frames without duplicate removal or other engineering tricks, the total number of symbolic compositions will be $N^T$. To reduce the size of the search space, we primarily focus on single-frame policy distillation, i.e., infer the action only based on the current frame observation. For the environments that require multi-frame tracking, those

long-term informations could be attached to the object as an attributes of the object, and are carried and updated across time. For example, in the shooting based environments, whether an object has been set as the "next shooting target" by the teacher agent could be set as an attribute of this object, and this attribute could be measured by observing the teacher behavior, and backtrack in time to mark previous frames (backtrack is allowed during distillation, since our distillation algorithm works with the offline stored teacher behavior dataset), and could be presented as a component in the measured numerical state. Under this strategy, the size of the symbolic search space is greatly reduced, and a plenty of environments are found to be viable for symbolic distillation (approximately 85% of 1000+ gym retro environments). A sampled list of such environments are put in Appendix D.

$$\texttt{policy-CircusCharlie} = (10 < x_{pot} - x_{ring} < 50) \times \texttt{go\_left} +$$
$$\neg(10 < x_{pot} - x_{ring} < 50) \times [(x_{ring} - x_0 < 40) \times \texttt{jump} + \quad (1)$$
$$\neg(x_{ring} - x_0 < 40) \times \texttt{go\_right}]$$

$$\texttt{policy-pong} = (x_{\text{pong}} > 40) \times ((y_{pong} > y_S) \times \texttt{down} + \neg(y_{pong} > y_S) \times \texttt{up}) +$$
$$\neg(x_{\text{pong}} > 40) \times \texttt{no\_action} \quad (2)$$

### 4.3 ROUNDTOURMIX: THE PROCEDURE TO DISTILL THE SYMBOLIC POLICY

Having established the search space and clarified the learning targets, in this section, we propose the RoundTourMix: a gradient-free policy distillation approach that iteratively optimize the three learning target discussed in Section 4.1, by switching back and forth between observing teacher's behavior and interacting with the environments. It follows the straightforward philosophy of genetic programming: a mainline guess of symbolic policy tree is maintained, new variants are repeatedly mutated from the mainline, and those better variants which minimize the loss with the teacher's actions are captured. We take the Cross-Entropy loss as the metric under the discrete action space we tested. We also note the choice to use Mean Squared Error loss if the action space is continuous. Through this evolution procedure, the more complex and better performing mutations of symbolic policy are iteratively discovered.

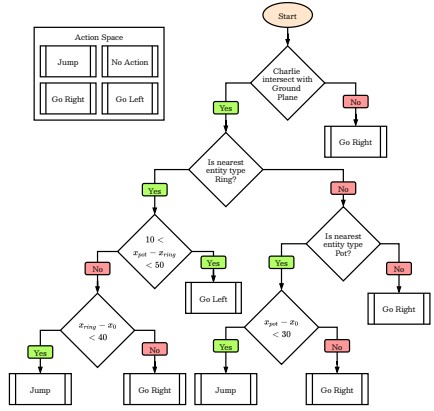

Figure 3: One sample distilled policy from a CNN agent in the CircusCharlie environment.

We note that the decision procedure of a wide range of CNN policy networks could be efficiently represented as high-fidelity tree shaped symbolic policy similar to Fig. 3. Therefore, we extend it into a generalizable form, where there are *condition node* and the *action node* co-existed. The condition node serves as the branching node, which corresponds to the learning target ❷, and the action node serves as the leaf node that corresponds to the learning target ❸.

The *condition node* has three key properties: the *condition*, $a_{LEFT}$, and $a_{RIGHT}$. Given a sample of numerical state, the *condition* is a symbolic composition that evaluates into a boolean value. $a_{LEFT}$ and $a_{RIGHT}$ are two of its children tree nodes, which can be either condition nodes or action nodes. Given a numerical state, if the *condition* of this node is evaluated to be True, $a_{LEFT}$ will be executed, otherwise $a_{RIGHT}$ will be executed.

The *action node* also have three key properties: the *total condition*, the *depth*, and the *policy*. The *total condition* is the cascaded condition of its ancestral condition nodes, in other words, given any sample state, the *total condition* of an action node is true only if the *condition* of all of its ancestral nodes are met (either True or False), so that this action node can be reached along the path from the root node. The *depth* of the action node means the length of the path from the root node to the action node. The *policy* can either be a single action, or a mapping from the state space (a subspace that already meets the *total condition*) to the action space. In practice, we encourage the *policy* to be a single action, but if the tree search reached certain predefined maximum depth, we also allow this *policy* to be the symbolic mapping recovered by symbolic regression (Cranmer et al., 2020; Cranmer, 2020), which is a subtree itself.

Specifically, the RoundTourMix is composed of four mutually interactive stages:

**1. Record Teacher Behavior.** In the first stage, the teacher interacts with the environment, and the visual observations are fully recorded and converted to geometric symbols frame-by-frame. We align these frames of symbols with actions and store as an *offline teacher behavior dataset* $\mathbf{D_0}$ (no need to store teacher's reward). This stage serve as the initialization of RoundTourMix, and is the only stage that will be executed only once.

**2. Guess and Observe.** In this stage, we make a random guess for the target geometric relations to be measured, then replay the teacher's interactions in $\mathbf{D_0}$. During the replay, the numerical states are evaluated using geometric operators, and are recorded (instead of measure all possible geometric relations and yield gigantic numerical states). We collect the numerical states and teacher's actions as a temporary dataset $\mathcal{D}$, and split it into a training set $\mathcal{D}_{train}$ and a validation set $\mathcal{D}_{val}$.

**3. Solve.** In this stage, we iteratively search the symbolic policy tree composed of the numerical operators. Table 1 presents the details of one round symbolic tree generation. This procedure is repeated for multiple times, after which a few of the best policy candidates on $\mathcal{D}_{val}$ are collected and sent to the next stage.

**4. Verify and Optimize.** In this stage, we let go the best few symbolic policies collected from last stage to interact with the environment. If any candidate symbolic policy is found to match or surpass the teacher's score, the RoundTourMix procedure is terminated and the distilled symbolic policy is found. Otherwise, return to the **Guess and Observe** stage and make new guess of which geometric relations to measure.

---

**Algorithm: Distilling Teacher Behavior into Symbolic Tree**

---

**Require:** Temporary dataset $\mathcal{D}_{train}$ containing $\mathbf{X}$ (numerical states), $\mathbf{Y}$ (actions)
**Return:** $r$: the root of symbolic policy tree
**Maintain:** $\mathcal{S}$: the set of unsolved action nodes

1: **Initializations**
2:     $r \leftarrow newActionNode(depth = 0)$
3:     $\mathcal{S} \leftarrow \{r\}$ ; $cnt \leftarrow 0$
4: **While** $\mathcal{S} \neq \{\}$ & $cnt < cnt_{MAX}$:
5:     $cnt \leftarrow cnt + 1$
6:     $n \leftarrow pop(\mathcal{S})$    ▷ Sample action node
7:     $\mathbf{Y}_{sub} \leftarrow \mathbf{Y}[n.\text{total\_condition}]$ ▷ Slices
8:     IF $Entropy(\mathbf{Y}_{sub}) < \Theta_{entropy}$:
9:       $n.\text{policy} \leftarrow Mean(\mathbf{Y}_{sub})$
10:   ELSE:      ▷ Single action cannot fit
11:      IF $n.\text{depth} < depth_{MAX}$:
12:        With probability $p_1$: ▷ Split condition
13:         $n \leftarrow newConditionNode()$
14:         $\mathcal{S} \leftarrow \mathcal{S} + \{n.a_{LEFT}, n.a_{RIGHT}\}$
15:        With probability $1 - p_1$: ▷ De-noise
16:         $n.\text{policy} \leftarrow$ default action
17:      ELSE: ▷ Too deep, stop branching further
18:        With probability $p_2$:
19:         $\mathbf{X}_{sub} \leftarrow \mathbf{X}[n.\text{total\_condition}]$
20:         $n.\text{policy} \leftarrow runSR(\mathbf{X}_{sub}, \mathbf{Y}_{sub})$
21:        With probability $p_3$:     ▷ De-noise
22:         $n.\text{policy} \leftarrow$ default action
23:        With probability $1 - p_2 - p_3$:
24:         $n' \leftarrow Sample(pathToRoot(n))$
25:         $removeSubtree(n')$
26:         $n' \leftarrow newConditionNode()$
27:         $\mathcal{S} \leftarrow \mathcal{S} + \{n'.a_{LEFT}, n'.a_{RIGHT}\}$
28: **Return** $r$

---

Table 1: RoundTourMix's solve stage algorithm.

**Remark 1: More Details in Stage 3.** The core part of the algorithm in Table. 1 is to repeatedly guess branching conditions of the *condition nodes* (lines 13/26), slice out the subset of data that obey the *total condition* (line 7), and fit action nodes with this subset of data (lines 9/16/20/22). The more accurate the guessed condition, the easier it is to fit action node. If the guessed condition happen to be near exact, the sliced teacher's action tends to be nearly deterministic (i.e., low entropy, line 8, where $\Theta_{entropy}$ is a threshold of the entropy). Otherwise, if the maximum depth $depth_{MAX}$ is met and the leaf node still is not deterministic (line 17), then it has the option to directly fit symbolic mapping (Cranmer, 2020) (line 20).

**Remark 2: Teaching Behavior Denoising.** An important mechanism in this algorithm is the *denoising* for the teacher behavior (lines 9/16/22). Under some conditions, the teacher model could possibly take noisy or "seemingly random" actions, which make it difficult for the RoundTourMix to find a single deterministic action that matches well (line 8 not satisfied). In these cases, the distillation algorithm has the chance to assign the default action (line 16,22) to *denoise* it. On other scenarios, the actions subset $\mathbf{Y}_{sub}$ under certain *total condition* have low entropy but are not entirely deterministic, the symbolic distillation can still *smooth* it into a deterministic single action (line 9), if the entropy is below threshold $\Theta_{entropy}$. Eventually, with the aid of the last screening stage of RoundTourMix, the symbolic policy taking fewer but more efficient actions are picked as the winner.

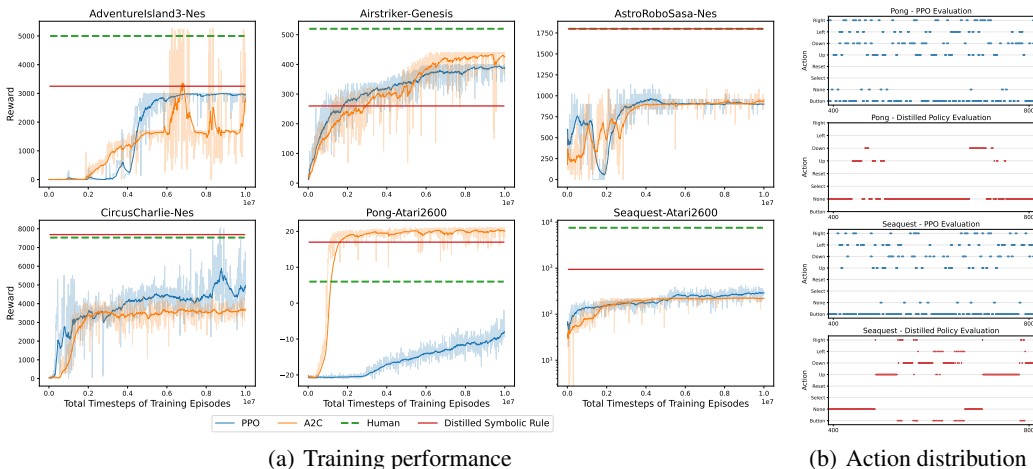

(a) Training performance          (b) Action distribution

Figure 4: Performance validation and behavior comparisons for the CNN based RL agents and the distilled symbolic rule. Action distribution (right) is for a single evaluation round

Although this *smoothing and denoising* effect might potentially overlook some subtle behaviors of the teacher agent, we argue that the benefits of introducing such mechanism outweigh the disadvantages. On one hand, symbolic rules can still be encouraged to evolve into more complex trees to capture intricate teacher behaviors, e.g., by setting the depth of the tree controllable by $depth_{MAX}$, and the richness/complexity of the symbolic building blocks, as defined by the search space. On the other hand, not all environments require intricate manipulations. For a few environments that challenges on "balancing ability" (no need to involve much reasoning), such as CartPole or LunarLander, it is difficult to distill reliable symbolic policy. However, the visual reasoning tasks dominates the gym retro environments, and in these tasks, the denoised symbolic rules have demonstrated quite unbeatable competence across a host of them, as to be seen in section 5.

## 5    EXPERIMENTAL SETTINGS AND RESULTS

In this section, we select several retro visual RL environments to train CNN policy networks, then distill, verify and interpret the symbolic policy. Specifically, the selected environments include: AdventureIsland2, AdventureIsland3, Pong, CircusCharlie, Airstriker-Genesis, Seaquest, and AstroRoboSasa. More illustrative visualizations of our experiments can be found in the videos in the supplementary materials, and we sincerely recommend our readers could check them out.

**Performance validation**. We first compare the performance of the distilled symbolic policy against the RL teacher models. Fig. 4 shows the performance of PPO (Schulman et al., 2017), A2C (Mnih et al., 2016), human player, and the proposed symbolic policy distillation. In the experiments, the symbolic policies are all distilled from the PPO teacher model. The score of the human player is obtained by a gaming master controlling the agent under the lowered frame rate. Due to engineering facilitation considerations, in this work we set the agents in the environments as to only have one life for all. As can be seen in the figure, the distilled policy maintained comparable performance with the teacher model, and for some environments, the distilled policy is even better thanks to the denoising mechanism of symbolic distillation.

**Interpretability validation**. Besides verifying the performance, we also evaluate the interpretability of the regressed policy. This requires to check the symbolic operators combinations learned in the tree nodes, re-write them as readable form and analyze. On the selected environments, highly interpretable and hierarchical forms of symbolic policy trees are all found in the distilled equations. One sample distilled policy for Pong is displayed in Eq. 2, and one sample for CircusCharlie is displayed in Fig. 3. More interpretable distilled policies on other environments are put in appendix C.

**Behavior comparison with the neural network teacher**. In Fig. 4, we plotted the action and the accumulated reward, both for the trained teacher model and the distilled symbolic policy. As can be seen in the figure, a well-trained teacher model make more frequent actions, while still underperform

| Evaluation on | AdventureIsland3 | AdventureIsland2 | | | | | |
|---|---|---|---|---|---|---|---|
| Checkpoints from AI-3 | 10M | 1.0M | 2.5M | 5.0M | 7.5M | 10M | Best |
| A2C | **5150** | 0 | 200 | 50 | 100 | 100 | 200 |
| PPO | 3000 | 150 | 0 | 100 | 50 | 50 | 50 |
| Distilled Symbolic Policy | 3250 | **1950** | | | | | |

Table 2: Policy Transfer comparisons. PPO and A2C are trained on AdventureIsland3 (AI-3), tested on AdventureIsland2. The symbolic policy is distilled from PPO agent trained on AdventureIsland3. Neither PPO/A2C nor symbolic rules are fine-tuned/modified on AdventureIsland2.

the distilled symbolic rule. More differences are better shown in the video in the supplementary materials. The results there shows that in the Pong env, the trained CNN-based teacher model frequently "jitters" up and down even when the pong is very far away from the right racket. In those cases, the pong is usually going to bounce with the wall to change its heading direction, making those states "hardly controllable". Due to the near-random behavior of the teacher under these "difficult" states, the symbolic distillation fails to converge to one single action under these cases, and hence mapped default action to them, accidentally generated more efficient rules, and those rules are picked through the last screening step of RoundTourMix.

**Transferablity validation**. Since the object detection step provide robust environment representations that group different scenarios together, even if the scenes' pixel-level attributes are markedly different, the distilled symbolic rule should transfer well if the logical specifications are congruent. In order to test the transferability/generalizability of the distilled symbolic rule, we compare the the distilled rules against CNN policies using the AdventureIsland2 and AdventureIsland3 environments (Refer Fig. 5). We first train a teacher CNN policy networks on AdventureIsland3 and distill symbolic policy with PPO; we then directly test both the CNN models and the distilled symbolic rule on AdventureIsland2, without tuning the CNN policy networks nor changing the symbolic rule.

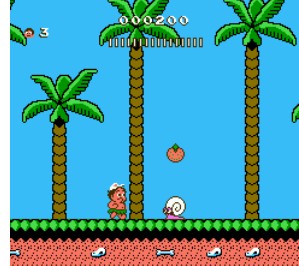

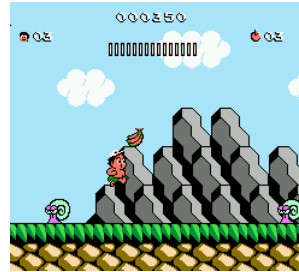

As can be seen in Table 2, when directly transfered to a new environment with similar underlying rule but slightly different image styles, the CNN models completely fail to match their performances in the original environment, over the checkpoint from the entire training history. Additionally, fine-tuning the CNN policy network to adapt to the new environment will require another considerable amount of training. In contrary, the distilled symbolic rule painlessly transfer to the new environment, with significantly fewer performance loss. This is due to that our object detection module dwells on optical flow and object shape, which is robust to color/style changes. The comparison between the symbolic policy and the CNN based policy network again endorses the better transferability of the symbolic rule.

Figure 5: Visual comparison between the AdventureIsland2 (top) and AdventureIsland3 environments (bottom)

## 6 CONCLUSION

This paper study the distillation of CNN based reinforcement learning agent into a symbolic policy that dwells on geometric and numerical operands and operators. Thanks to the the denoising and the screening mechanisms in the distillation procedure, the distilled symbolic policy achieves comparable or even better performance than the CNN teacher model. Our results point to the new opportunity to make the visual RL more interpretable, reliable and generalizable, by reclaiming a symbolic design with the proposed symbolic policy distillation. Our future work aims for more integrated neural-symbolic solutions for visual RL.

## 7 REPRODUCIBILITY CHECKLIST

To ensure reproducibility, we use the Machine Learning Reproducibility Checklist v2.0, Apr. 7 2020 (Pineau et al., 2021).

- For all **models** and **algorithms** presented,
  - **A clear description of the mathematical settings, algorithm, and/or model.** We clearly describe all of the settings, formulations, and algorithms in Section 4.3.
  - **A clear explanation of any assumptions.** We do not make assumptions.
  - **An analysis of the complexity (time, space, sample size) of any algorithm.** We do not make the analysis.
- For any **theoretical claim**,
  - **A clear statement of the claim.** We do not make theoretical claims.
  - **A complete proof of the claim.** We do not make theoretical claims.
- For all **datasets** used, check if you include:
  - **An explanation of any data that were excluded, and all pre-processing step.** We did not exclude any data or perform any pre-processing.
  - **A link to a downloadable version of the dataset or simulation environment.** All ROMs used can be downloaded from `https://archive.org/details/No-Intro-Collection_2016-01-03_Fixed` and `http://www.atarimania.com/rom_collection_archive_atari_2600_roms.html`.
  - **For new data collected, a complete description of the data collection process, such as instructions to annotators and methods for quality control.** We do not collect or release new datasets.
- For all reported **experimental results**, check if you include:
  - **A clear definition of the specific measure or statistics used to report results.** We use the accumulated reward gained in a single episode as our statistical measure.
  - **A description of results with central tendency (e.g. mean) & variation (e.g. error bars).** We do not report mean and standard deviation for experiments.
  - **The average runtime for each result, or estimated energy cost.** We do not report the running time or energy cost.
  - **A description of the computing infrastructure used.** We do not report the computing infrastructure used.

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

## A  JUSTIFICATION FOR KEY NOTES

In this section we provide justifications for the architectural and algorithmic designs of our proposed RoundTourMix algorithm.

### A.1  DETAILS ON THE SEARCH SPACE

**Geometric and Numerical Operator Search Space.** Precisely, we have a total of 11 different functions in the geometric operator search space and 19 different functions in the numerical operator search space. As mentioned in section

**Temporal Tracking and Memory-Attributes.** We include mechanisms that make our symbolic algorithm aware of the temporal behavior of its entities using memory-attributes. For instance, in shooting-based environments, relying just on the current frame gives rise to the agent shooting multiple times at the same enemy if it is not aware of the enemies that it has already launched its bullet at. Another memory-attribute in such games arises when the agent needs to remember the target (out of multiple targets) it has locked on to, get in a suitable position and then fire.

### A.2  MOTIVATIONS BEHIND THE ALGORITHMIC CHOICES

In this section we break down the Step 2 and Step 3 of RoundTourMix while providing validation for the specific choices in the same.

**Justification for Random Selection.** Exhaustive search will ensure that excellent performance will be achieved with virtually unlimited hardware and software resources. However, as previously mentioned in 4.2, one potential challenge of the symbolic distillation technique is that the geometric representation's search space is exponentially huge. As a result, as compared to exhaustive search, random selection is the only viable option. Furthermore, our investigations show that the random search paradigm produces more reliable outcomes over exhaustive searching. If we compose symbolic compositions of length 20, the total possible policies are $(11 + 19)^{20} = 30^{20} = 3.48e29$ which is a gigantic number. Furthermore, because there is no "learning" involved, any exhaustive or brute-force search goes against the spirit of reinforcement learning. Traditional reinforcement learning does not use such comprehensive methods because of the vast number of viable policies, as earlier noted.

**Guess and Observe.** As earlier discussed the geometric search space is very large. Hence we randomly guess the 16 different geometric relations to be guessed. We then iterate over the image from the teacher behaviour, evaluate the randomly sampled geometric and numerical relations. For each frame we create a vector of size 16 of possible logics which are then later on used as the numerical state for symbolic distillation - The set numerical state vectors as extracted from each image constitutes the offline teacher dataset

**Explanation of Branching Conditions.**

- Search the symbolic policy tree composed of numerical operators
- Performed to get suitable candidate policies
- We perform the search a total of $cnt_{MAX}$ times
- At the start of each search we first sample an action node from the set of all unsolved action nodes
- We then slice out the subset of the offline teacher data that aligns with the guessed branching condition.
- If the sampled branching condition was a step in the right direction, the entropy of the subset sliced remains at a low value
- If it does not, it is an indicator that the underlying rule was more complex and the path is to be explored deeper
- If the tree is already at its maximal depth, we either (A) directly run and fit symbolic regression between the sliced numerical state vs action offline dataset (B) assign a default action to that action node or (C) prune off the entire sub-tree containing the action node in question
- Else if the tree is not yet at its maximal depth and the entropy is high, we either (A) explore deeper by creating a left/right branching condition or (B) assign a default action at the action node in order to denoise it

## B    DETAILS OF OPTICAL FLOW BASED OBJECT DETECTION

Since in most retro environments, the image styles are simple, where the backgrounds are nearly fixed and the objects share similar patterns. Therefore, we take the optical flow as a simplified object detection module.

We first segment out the object list from template matching, then estimate the velocity using the pretrained FlowNet (Ilg et al., 2017), and get the speed via the average optical flow of detected objects. The classes of the objects are obtained based on the shape in the predefined templates.

## C    EXAMPLES OF DISTILLED SYMBOLIC POLICY

We check the learned symbolic operators in the tree nodes, and re-write them as interpretable symbolic tree policy. The results are plotted in Fig. 9, Fig. 7, Fig 10, Fig 6, Fig 8, and Fig 11.

## D    EXAMPLES OF RETRO-ENV LIST THAT ONE-FRAME SYMBOLIC DISTILLATION CAN WORK WELL

After checking the underlying logics and evaluated the possible strategies of gym-retro environments, a plenty of them are found to be solvable by our symbolic policy distillation. Several examples include: 8Eyes-Nes, 1942-Nes, BadDudes-Nes, BioSenshiDanIncreaserTonoTatakai-Nes, BreakThru-Nes, BubbleBobble-Nes, CodeNameViper-Nes, FrontLine-Nes, GhostsnGoblins-Nes, KidIcarus-Nes, KidNikiRadicalNinja-Nes, Parodius-Nes, RoboccoWars-Nes, Sansuu5And6NenKeisanGame-Nes, Seicross-Nes, SonSon-Nes, Tennis-Atari2600, TwinEagle-Nes, ViceProjectDoom-Nes, Xexyz-Nes, MysteryQuest-Nes. The screenshots for them are displayed in Fig. 12.

## E    OTHER ILLUSTRATIONS

The pseudo-code for the algorithm in table  1 are presented in Fig. 13

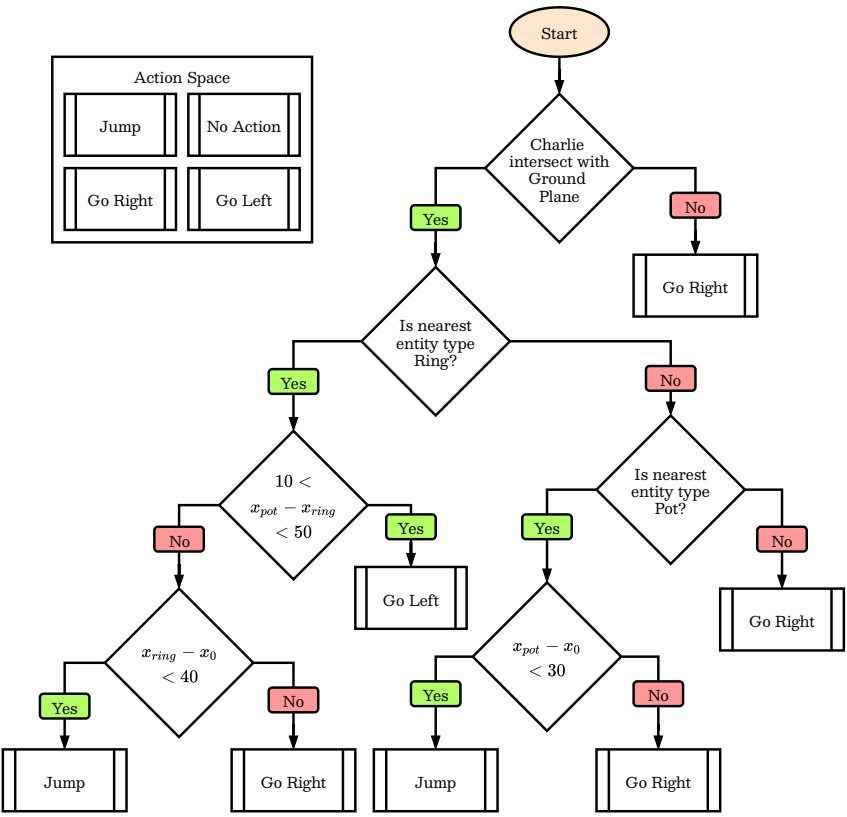

Figure 6: Distilled policy for CircusCharlie-Nes

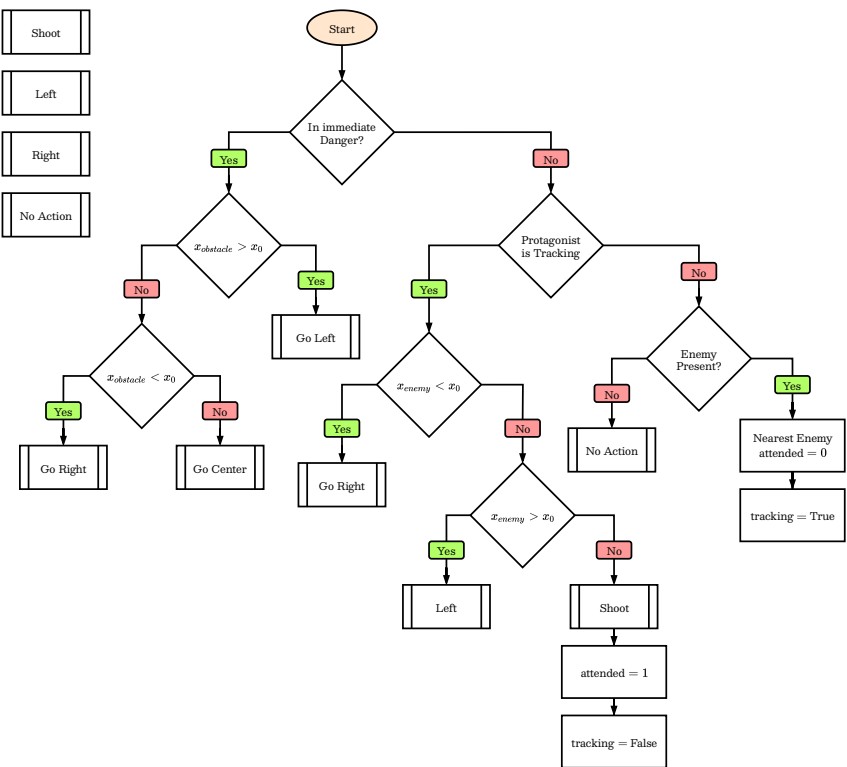

Figure 7: Distilled policy for Airstriker-Genesis

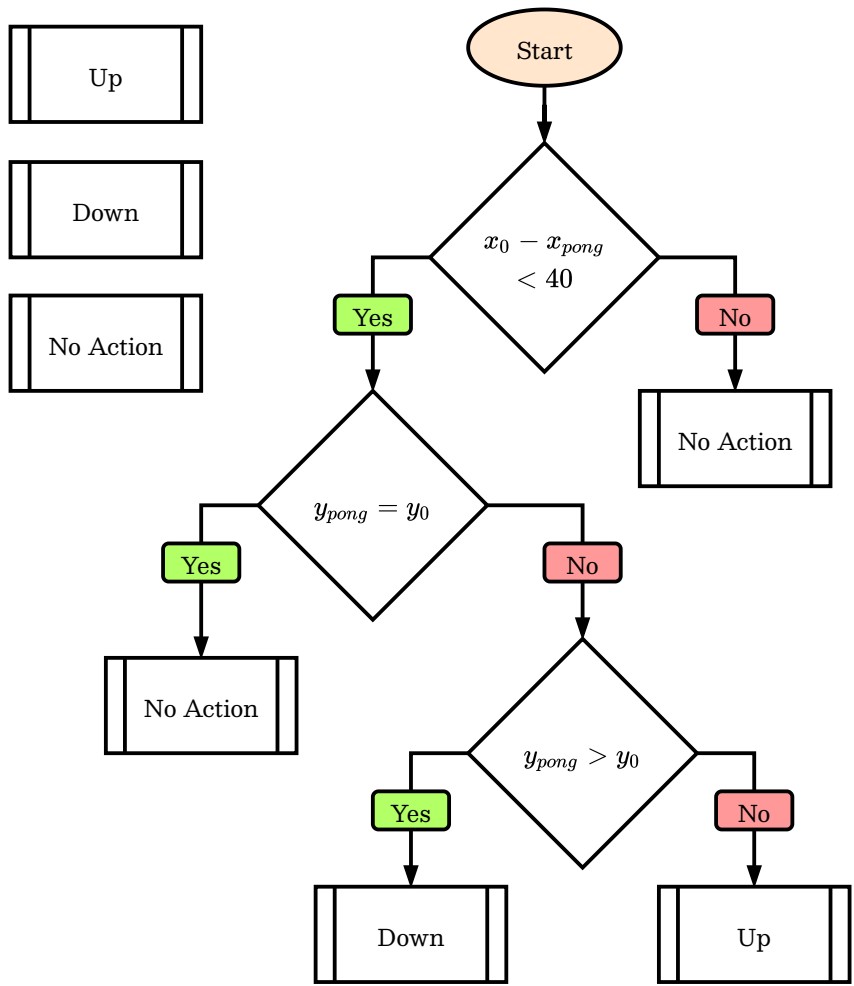

Figure 8: Distilled policy for Pong-Atari2600

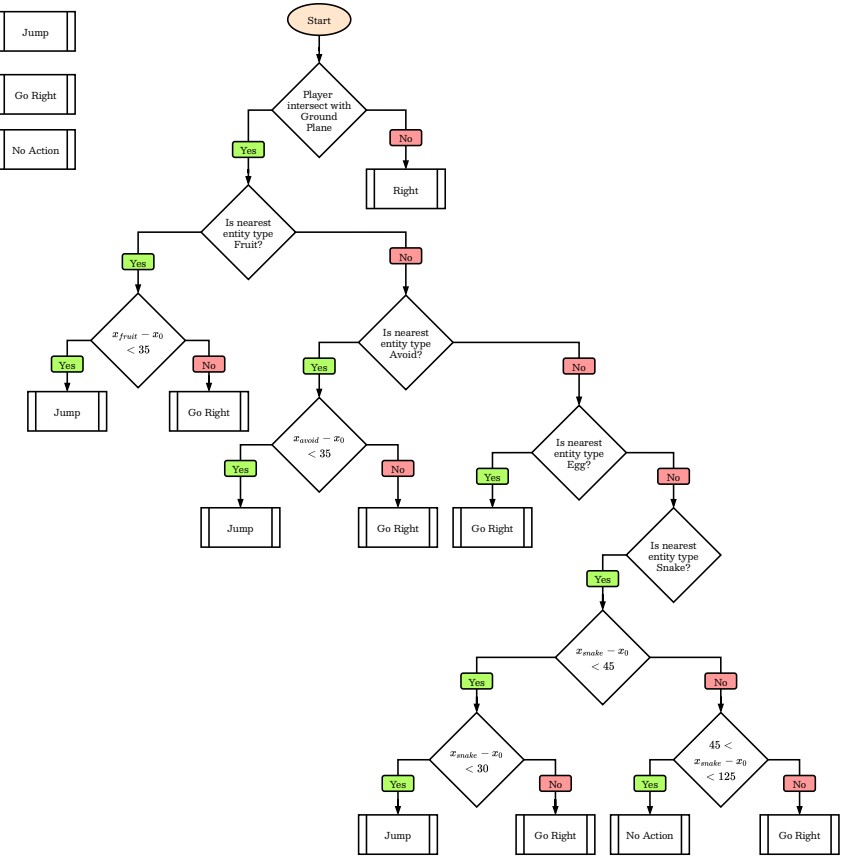

Figure 9: Distilled policy for AdventureIsland3-Nes

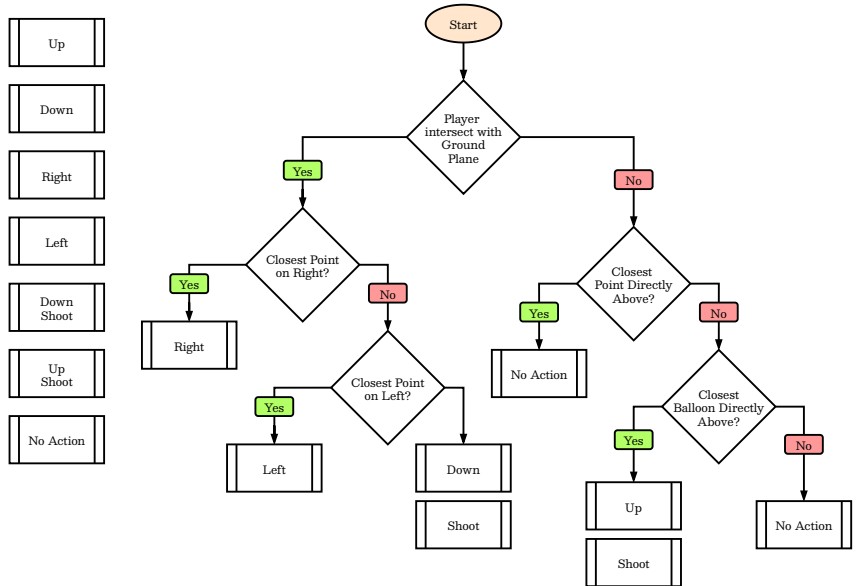

Figure 10: Distilled policy for AstroRoboSasa-Nes

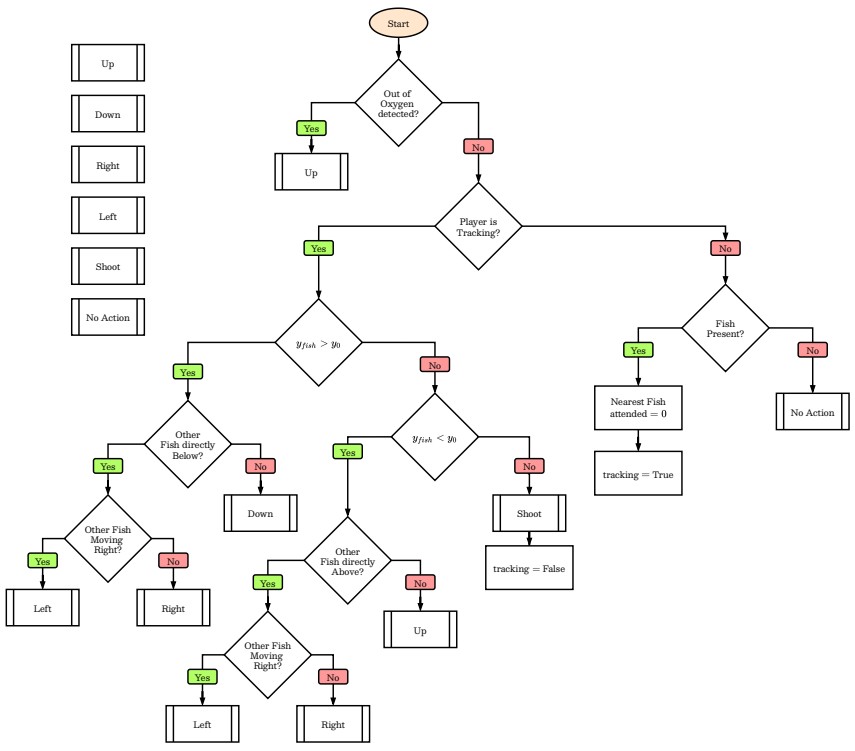

Figure 11: Distilled policy for Seaquest-Atari2600

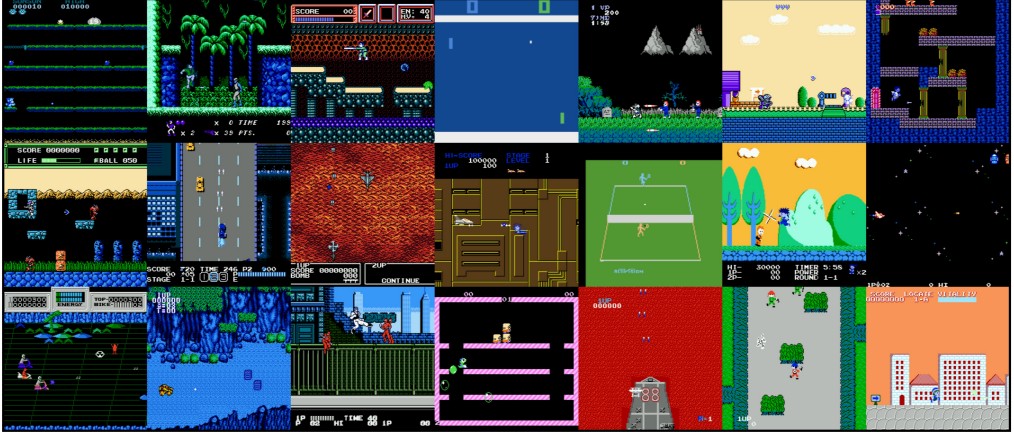

Figure 12: Several environments found to be viable for symbolic distillation.

```
# psudo-code for the solve stage of RoundTourMix
def solve_policy_as_symbolic_tree(x, y):
    # input is a list of pairs of teacher behaviors:
        # x: numerical state
        # y: action
    # output: a symbolic tree with condition nodes and action nodes
    root = new_action_node(depth=0) # initialize the root node as an action node
    unsolved_action_nodes = { root }
    loop_cnt = 0
    while (unsolved_action_nodes is not empty) and (loop_cnt < max_cnt):
        loop_cnt += 1
        node = sample(unsolved_action_nodes).pop() # randomly sample an unsolved action node
        # First check if the actions under the current total_condition is near deterministic.
        y_subset = y[node.total_condition] # select slices that satisfy total_condition
        if entropy(y_subset) < entropy_threshold:
            # If a single action fits under the current total_condition, then resolve and close this branch
            node.policy = mean(y_subset)
        else:
            if node.depth < max_depth:
                # If max depth is not met, branch on this node by a randomly guessed
                # condition, and mark new child nodes as unsolved
                replace_action_node_with_new_condition_node(node)
                unsolved_action_nodes.add([node.a_LEFT,node.a_RIGHT])
            else:
                # If the current node is already too deep, then stop branching further.
                uniform_0_1 = rand() # sample from a uniform distribtion [0,1]
                if uniform_0_1 > p_SR:
                    # With probability p_SR, directly solve this node using Symbolic_Regression.
                    x_subset = x[node.total_condition]
                    node.policy = Symbolic_Regression(x_subset, y_subset)
                elif uniform_0_1 > p_SR + p_default_action:
                    # With probability p_default_action, set to default action to de-noise teacher behavior.
                    node.policy = default_action
                else:
                    # Otherwise, remove a subtree containing  this node, then renew the searches.
                    node_father = sample(node.father_nodes_list)
                    remove_subtree(node_father)
                    node_father = new_condition_node()
                    unsolved_action_nodes.add([node_father.a_LEFT,node_father.a_RIGHT])
    return root
```

Figure 13: The pseudo-code for the algorithm in table 1.

