# OpenReview forum: "Reasoning With Hierarchical Symbols: Reclaiming Symbolic Policies For Visual Reinforcement Learning"
_ICLR.cc/2022/Conference — ICLR 2022 Submitted_

### Official Review · Reviewer_GGAU · 2021-10-28

**Correctness:** 4
**Technical Novelty And Significance:** 3
**Empirical Novelty And Significance:** 2
**Recommendation:** 6
**Confidence:** 4

**Main Review:**

Strengths:

- To my knowledge this is a novel algorithm that shows promising results on a small number of games and shows to be more robust to changes in the environment.
- Resulting decision making is easily interpretable by humans
- The work is very clearly presented

Weaknesses
- Minor: “Abstracting pure symbolic rules from data is not new. The family of symbolic regression (SR) methods (Cranmer et al., 2020; Runarsson & Jonsson, 2000; Gustafson et al., 2005; Orchard & Wang, 2016) have been studied to to directly.” Remove double “to”.
- Being a purely offline learning method it seems like the maximum possible performance of this method is closely related to the expert policy and the types of levels that training experience is gathered on. It would be very interesting to know whether the authors have thought about whether their proposed method could be extended to learn from online data generated by itself.
- Another concern is that thinking about their proposed method, it seems that the method thrives on levels where the optimal action can be inferred from the current observation and that a predominantly reactive policy can perform well. I am curious about how their method would perform in a game like sokoban that requires careful planning, or others, where the agent needs to remember information that is no longer visible in the current observation. It seems to me that their proposed symbolic method does not have any memory and would therefore not achieve similarly impressive results.


**Summary Of The Paper:**

The paper presents a novel algorithm that distills expert experience in the Gym Retro Games environment into an interpretable and symbolic policy. The main steps in their method are the following: first gather experience from a trained network and pre-process with pre-trained and frozen networks to extract a list of objects, their type, position and velocity for every image observation. Then these are passed to the next module that computes numerical states, such as auxiliary lines between the objects and computing relations between the objects. Finally, their novel algorithm makes use of genetic mutation to discover rules that consume these relations and take actions. The last part is trained to match the data from the expert and only then deployed for evaluation in the environment. Their empirical results show that their method can perform well and even outperform the expert policy used to collect the initial experience. Moreover, they show that their proposed method transfers much better compared to the end-to-end baselines (PPO and A2C) from AdventureIsland3 to AdventureIsland2, making it a more robust method. Importantly, they show how easy it is to interpret the decision making that selects the action from the current state.

**Summary Of The Review:**

I recommend a weak acceptance for the paper, due to the novelty of their algorithm and the interesting empirical results presented, where the final policy is very interpretable. My main concerns are with the wider applicability of the method.

---

> ### Author Response · Authors · 2021-11-18
> **Authors' response to reviewer GGAU**
>
> Dear reviewer GGAU:
>
> We appreciate your acknowledgement of our work and your interesting ideas in your comments! We have answered all of your questions as follows, and hope they help to better understand the proposed method.
>
> **Q1:** Remove the double “to”.
>
> **Reply:**
>
> Thanks for your carefulness in spotting it. We’ve made the changes accordingly.
>
>
> **Q2:** It would be very interesting to know whether the authors have thought about whether their proposed method could be extended to learn from online data generated by itself.
>
> **Reply:**
>
> This is a good question. There are two components in the distilled equations, the symbolic skeletons (operator/operand compositions), and the coefficients in the symbolic skeleton. One can make the symbolic rules amadendable to online learning by considering modifying these two components. An easy setting for online adaptation is to tune the coefficients: they are all trainable parameters, and are able to receive gradients just as the weights in the CNN rule do. For example, in the decision node $x_{protagonist} - x_{pot}<50$, one can set the 50 to be a trainable weight $w$ and use a traditional RL approach to tune this parameter.
>
> The symbolic skeleton is more difficult to learn than its coefficients. One way of learning it is first randomly mutating certain components, then fine-tune the newly introduced coefficients to see if better performance can be achieved, otherwise discard the mutation. Another possible yet more difficult approach is to train a sequential decision maker (RNN or GPT) to output the replacement symbols. Indeed we are actively extending this work by considering the second direction, to train a GPT to learn better decision rules. We also refer to several relevant papers as follows.
>
> [1] Sheikh, Hassam, et al. "Learning intrinsic symbolic rewards in reinforcement learning." arXiv preprint arXiv:2010.03694 (2020).
>
> [2] Sieusahai, Alexander, and Matthew Guzdial. "Explaining Deep Reinforcement Learning Agents In The Atari Domain through a Surrogate Model." Proceedings of the AAAI Conference on Artificial Intelligence and Interactive Digital Entertainment. Vol. 17. No. 1. 2021.
>
> [3] Landajuela, Mikel, et al. "Discovering symbolic policies with deep reinforcement learning." International Conference on Machine Learning. PMLR, 2021.
>
> [4] Kubalík, Jiří, et al. "Symbolic regression methods for reinforcement learning." IEEE Access (2021).)
>
>
> **Q3:** I am curious about how their method would perform in a game that requires careful planning where the agent needs to remember information
>
> **Reply:**
>
> This is a good question. In fact the symbolic form does allow long-term memory to be retained, and we’ve provided examples for it. The long term memory can be captured as the attributes of the detected objects, which can be maintained overtime (stated in the last paragraph of section 4.2). As an example of long term behavior, figure 11 shows in the distilled rules of Seaquest, where “player is tracking” means the player is consistently focused on the current object. Another similar case is in Fig 7 for Airstriker-Genesis.

---

> > ### Comment · Reviewer_GGAU · 2021-11-19
> > **Thank you for your response**
> >
> > I would like to thank the Authors for their response to my questions.
> >
> > With regards to your reply to my Q2, I look forward to seeing this implemented in future work and I am curious to see how well this will perform in practise.
> >
> > Regarding my Q3, thank you for pointing out that it's explained at the end of section 4.2, I somehow seem to have missed it. Thank you for explaining. Am I right in understanding that this requires the researcher to a priori encode all possible attributes that are necessary to solve all these games? for example, if the game required the agent to pickup a key from one room to open the door in another, the researcher would have to encode a "holds_key" attribute, correct?
> >
> > Moreover, I am not sure I fully understand how the "is tracking" node is decided. What would a heuristic look like to resolve that?
> >
> > I also noticed some typos when reading this paragraph:
> > "For the environments that require multi-frame tracking, those long-term informations could be attached to the object as an attributes of the object, and are carried and updated across time."
> > -> In my opinion, this should become: "For the environments that require multi-frame tracking, this long-term information could be attached to the object as additional attributes of the object, and carried and updated across time."

---

> > > ### Author Response · Authors · 2021-11-20
> > > **Thank you for your reply**
> > >
> > > Dear reviewer GGAU:
> > >
> > > Thanks for your reply.
> > >
> > > Regarding the necessity of defining possible attributes, it is necessary to define sufficient attributes to express the behaviors, but these attributes could be low level and *general purpose* attributes, instead of high level and cumbersome. In the "hold key" example, we could only define "A attached to B", and only after distillation, this A is found to always be the "key" object and B is found to be the protagonist.
> > >
> > > Regarding "how is the `is_tracking` node decided", we first set every objects to have this boolean attribute, learn to assign True or False values to these attributes. The objects hit by the agent is set to be True, while other objects are set to be False.
> > >
> > > We appreciate for pointing out this typo, and have changed the singular to the plural in our PDF. We hope our responses above help to further clarify our proposed methodology.
> > >
> > > Best,
> > >
> > > Authors

---

> > > > ### Comment · Reviewer_GGAU · 2021-11-29
> > > > **Thank you**
> > > >
> > > > I would like to thank the authors for their response.

---

### Official Review · Reviewer_AL2N · 2021-11-02

**Correctness:** 2
**Technical Novelty And Significance:** 2
**Empirical Novelty And Significance:** 2
**Recommendation:** 3
**Confidence:** 4

**Main Review:**

**Strengths**

* This paper is well-written and easy to follow. The proposed algorithm is also clearly explained and contains enough details to reproduce.
* To my knowledge, the proposed method is a novel approach compared to prior works. In Figure 4, the distilled symbolic rule recovers and even occasionally outperforms the original teacher NN policy, which demonstrates the effectiveness of RoundTourMix.
* I appreciate the increased interpretability of the distilled rules.

**Weaknesses**

While the results on Atari games and AdventureIsland are strong, I am primarily concerned with the applicability and effectiveness of the method on domains beyond toy 2D pixelated games. The paper makes many simplifying assumptions that may not hold in more complex and useful domains. To elaborate:

1. Section 3 discusses the "geometric symbol representation" of the game pixels, which requires a semantic parser to preprocess the image. The exact approach (described in Appendix A) is template matching + FlowNet, which overfits to the toy game domain. This can be very brittle and result in compounding error in the downstream rule selection. Even worse, it will not be compatible with more realistic domains, such as:

	* 3D environment where the embodied agent can move freely and change perspective. The simple geometric shape parsing will fail, as the background changes dramatically with egocentric motion.
	* Occlusion, overlapping objects, deformable objects, etc. The current symbolic representation cannot handle any of these complex scenarios.
	* Locomotion tasks with a complex robot morphology that is unclear how to parse from pixels.

Meanwhile, blackbox approach like PPO will be able to handle all of the above without the hand-engineered preprocessing pipeline.

2. Section 4.2 discusses the geometric operator search space, which contains elements like `velocity_extension` and `static_line_drawer`. These are hand-engineered primitives tailored for simple 2D environments that cannot be easily applied to more sophisticated tasks. For example, how to handle complex contact interactions between the robot and its surroundings? What about deformable objects? Objects that change state (like cooking tasks)? Navigation in 3D world? Mobile manipulation? Once again, approaches like PPO can adapt to all these tasks _without case-by-case modifications_. This severely limits the usefulness of the proposed symbolic rule system.

3. The symbolic rules do not seem to have any mechanism for long-term memory. How can it be applied to tasks that require long-horizon planning and reasoning, rather than short-term "muscle memory"? PPO + LSTM solves this out of box.

4. The "transferability validation" experiment in section 5 gives an unfair advantage to rule-based policy, because it makes the strong assumption that there exists a perfect object recognition system for all game levels, even with very different visual appearance. This certainly holds for AdventureIsland, but does not hold for any real-world tasks. The same invariance can be achieved for ConvNet if we pretrain the early layers of an NN policy to be domain-agnostic, using techniques like data augmentation, domain randomization, or contrastive representation.

5. The authors claim that "smoothing and denoising" help make the distilled rule more robust, and show the performance boost in Fig. 4 for certain tasks. However, similar "denoising" techniques can be applied to PPO as well, in the form of robust representation learning. For example, the following works all contribute to better neural representations and more robust policies. It is unclear if symbolic rules can consistently outperform these approaches:

    * Image Augmentation Is All You Need: Regularizing Deep Reinforcement Learning from Pixels. Yarats et al. ICLR 2021.
    * Reinforcement Learning with Augmented Data. Laskin et al. NeurIPS 2020.
    * CURL: Contrastive Unsupervised Representations for Reinforcement Learning. Srinivas et al.
    * Learning Invariant Representations for Reinforcement Learning without Reconstruction. Zhang et al. ICLR 2021.

Furthermore, I strongly disagree with the general claim that "our symbolic policy distillation approach captures the causal structure through data-driven experience" (introduction section).

First, the symbolic rule distills from an NN trained policy, which necessarily means that it will inherit any spurious correlation learned from the teacher. Even though the authors introduce tricks like "denoising", it will not be able to recover the true causal structure without any explicit mechanisms like do-calculus or counterfactual reasoning.

Second, the learned symbolic rules develop hard-coded constants that are _also very brittle_. For example, what if the Pong game is slightly stretched and the `X_pong` coordinate threshold is 45 instead of 40? Humans will have no problem generalizing, but the symbolic rule will fail even though the causal mechanism of the game remains the same (and visual appearance is almost identical).

**Summary Of The Paper:**

This paper introduces an algorithm called RoundTourMix to distill a trained neural network policy into an executable symbolic form. The algorithm is gradient-free and denoises the original end-to-end policy. Results are shown on Atari games and AdventureIslands, which demonstrate that the distilled symbolic rules retain the original performance level while being more resistant to domain shift.

**Summary Of The Review:**

While the paper is overall clear and easy to follow, it makes many strong assumptions that are tailor-made for simple 2D games. It is unclear how the symbolic rule framework can scale to more complex domains. I am unconvinced that the distilled rules are more robust or causal than the PPO counterpart, especially when trained with robust representation learning techniques.

I give my rating due to the limited usefulness and problematic assumptions of the proposed approach in realistic tasks.

---

> ### Author Response · Authors · 2021-11-18
> **Authors' response to reviewer AL2N (part 2)**
>
> Dear reviewer AL2N:
>
> Due to length limitation, we continue our response session here.
>
>
> **Q4:** The experiments pose an unfair advantage to rule-based policy, because it makes the strong assumption that there exists a perfect object recognition system - the same invariance can be achieved for CNN-based methods.
>
> **Reply:**
> First, we would like to note that our method doesn't make "rule versus CNN" rival each other. We start with a CNN policy and make it symbolic and interpretable, so we're compatible with any stronger CNN policy method.
>
> More importantly, stronger object recognition and stronger policy are two different things. We never claimed that our *object detection* step is *RULE-based*, instead, we *start* from the detected objects. Based on your statement "symbolic rule is not right since object detection is hard", we would like to again clarify the relation between object detection and symbolic rule: the latter does not include the former, and we only claimed the novelty of proposing the latter.
>
>
>
>
> **Q5:** Similar "denoising" techniques can be applied to PPO as well in the form of robust representation learning.
>
> **Reply:**
> We agree with this statement, however, it is irrelevant to the effectiveness of the proposed method. Our symbolic distillation method is compatible with stronger and more robust CNN policies.
>
> The general purpose of this paper is to distill the CNN based policy into a more interpretable symbolic form. In this sense, the obtained symbolic rules are surrogate models of the CNN based methods. The existing works you mentioned (which improved the robustness of neural representations) can be used as the teacher model for our white-box distiller, and these more robust teacher models would further ease the distillation procedure, and would lead to faster and smoother convergence.
>
>
> **Q6:** The reviewer strongly disagree with the claim "our symbolic policy distillation approach captures the causal structure through data-driven experience"
>
> **Reply:**
> Thanks for these comments. We originally meant to say symbolic distillation removes many nuisances by constraining the representation space. We now agree with you, and have rectified the correctness of our statement in the updated PDF.
>
> This statement has just appeared once in paper, and in no way it is our major claim. Removing it hardly hurts any contribution of our paper.
>
>
>
>
>
>
> We hope our answers help to clarify the true contribution and effectiveness of our proposed method, and lead to a fair and positive assessment.
>
> Best,
>
> Authors of paper 942

---

> > ### Comment · Reviewer_AL2N · 2021-11-22
> > **Further doubts on the rebuttal**
> >
> > Dear authors,
> >
> > Thank you so much for taking your time to write such an extensive and thoughtful response. I really appreciate your efforts.
> >
> > Similarly, I respectfully but strongly disagree with multiple points in your response.
> >
> > > *Furthermore, people can train a general-purpose CNN based object detector (out of many cheaply available yet provably robust benchmarks, such as Fast-RCNN/Yolo) and feed to our symbolic reasoning module without environment-by-environment training.*
> >
> > > *These geometric operators will receive formal definitions and rigorous computations in 3D cases, while preserving their general applicabilities. Such extension simply requires 3D vector computations instead of 2D*
> >
> > There is no **experimental evidence** presented in this paper to support the claim that your approach can scale to any of the more complex scenarios mentioned in my original review.
> >
> > To elaborate:
> >
> > First, adapting your method to realistic 3D environments is highly nontrivial and requires extensive changes that are not as simple as upgrading to an off-the-shelf 3D object detector. For example, if one wants to train a mobile manipulation robot in 3D embodied environments like Habitat [1] or AI2THOR [2], the background can change dramatically in egocentric view, and simply parsing the 3D objects is far from enough to handle the complex floor plans, room layouts, and rich object semantics. I do not see a straightforward way to apply rule-based policy on such agents, unless combined with more traditional techniques like SLAM, which significantly and unnecessarily complicates the framework. In contrast, PPO is able to handle these tasks with little modifications [3,4].
> >
> > * [1] Habitat: A Platform for Embodied AI Research. Savva et al.
> > * [2] AI2-THOR: An Interactive 3D Environment for Visual AI. Kolve et al.
> > * [3] Decentralized Distributed PPO: Solving PointGoal Navigation. Wijmans et al.
> > * [4] ManipulaTHOR: A Framework for Visual Object Manipulation. Ehsani et al.
> >
> > Second, you have not addressed how the method can work on more realistic object settings, like deformation or occlusion. For example, typical image segmentation models do not handle occlusion out of box, and one object will appear as 2 or more "broken" parts as input to the downstream symbolic rules. This can be a very common scenario in robotics, as the robot arm/hand itself moves and occludes part of the object. If the upstream detector makes systematic mistakes, the subsequent symbolic policy will also break down. These two parts cannot be cleanly separated.
> >
> > As for deformable objects (like sponge or towel), the proposed method does not seem able to handle warping object boundaries. The learned symbolic rules on Atari games showcased in the paper (Fig. 6 - 11 in Appendix E) only deal with simple intersections, centroid coordinate comparisons, and other simplistic geometric primitives.
> >
> > As for "complex robot morphology" (point 1c in my original review), let us simply take **Deepmind Control Suite** [4] as an example. This is one of the standard and most widely adopted image-based RL benchmarks. Variants of PPO and SAC have been extensively evaluated on DMControl, and have achieved impressive results [5,6]. **How can your proposed method handle this benchmark**? How do you parse the walker, cheetah, ant, humanoid robots, and build symbolic rules to control them to walk/stand/hop/run? Technically, all these robots are *a single object*, and it is unclear how to segment without hand engineering.
> >
> > While it may be possible to come up with different sets of primitives to address the above cases one by one, one would require huge amount of manual trial-and-error to select the best design. End-to-end approaches can be adapted to all these domains with little friction.
> >
> > * [4] DeepMind Control Suite. Tassa et al.
> > * [5] Image Augmentation Is All You Need: Regularizing Deep Reinforcement Learning from Pixels. Kostrikov et al.
> > * [6] Reinforcement Learning with Augmented Data. Laskin et al.
> >
> > ---
> > > *However, CNN based PPO algorithms have to be trained or fine-tuned for every single environment.*
> >
> > I want to point out that the proposed method *also requires* re-training or finetuning for most new tasks. You need a **CNN policy to distill from in the first place**. If the task semantics change, you will have to re-train the CNN policy anyway.
> >
> > Moreover, as I pointed out in the original review, **the learned symbolic rules are also very brittle because they produce hard-coded constants**. What if the Pong game is slightly stretched and the `X_pong` coordinate threshold is 45 instead of 40? The symbolic rules will struggle in such cases. The only scenario that the proposed method can generalize without re-training is (1) that the spatial locations and shapes of all objects stay the same, and (2) a perfect object detection system exists and is resistant to any visual appearance shifts. These are very strong assumptions that rarely hold in nontrivial tasks.

---

> > > ### Author Response · Authors · 2021-11-29
> > > **Responses to further concerns**
> > >
> > > Dear reviewer AL2N:
> > >
> > > We believe the points you raised should not be downgrading the merits of this paper.
> > >
> > > First for the comparison of CNN and symbolic rules transferrability, we have shown in the table 2 that the symbolic rule generalize, while the CNN does not. These cases correspond to when two scenarios have distribution shift, but the underlying logics are similar, which are usual cases. In these cases, researchers don't have to fine tune symbolic rule but need to fine tune the CNN.
> > >
> > > Second, the brittleness of symbolic constants does only hold if one restrict re-training it. However symbolic rules can be re-trained! If the Pong game is slightly stretched and the `X_pong` coordinate threshold is 45 instead of 40, one can still fine-tune the symbolic rule by optimizing the coefficients with gradient descent. Given the light weight property of symbolic rule as an advantage, such fine-tuning will be  faster than tuning CNN methods!
> > >
> > >
> > >
> > > Best,
> > >
> > > Authors of paper942

---

> ### Author Response · Authors · 2021-11-18
> **Authors' response to reviewer AL2N (part 1)**
>
> Dear reviewer AL2N:
>
> We appreciate all of your comments. We would like to respectfully point out your potential misunderstanding of our claim: the “*object detection method*” are not tied to the “*distilled policy*”, and the true contribution of this work is the latter not the former, a.k.a., we are the first to distill a CNN policy into a symbolic rule, and the “*object detection*” is only the input of the proposed method. Besides, we assumed a general OD algorithm which can be trained to fit to whatever environments needed.
>
> For all of your questions, we have addressed them and answered them as follows.
>
>
> **Q1:** The proposed method uses Template Matching + FlowNet which overfits on the Atari games domain.
>
> **Reply:** We agree that TM+FlowNet is limited to atari games, but our proposed framework is tied to general “object detection”, instead of TM+FlowNet: as stated in section3, TM+FlowNet is only the engineering shortcut. It is straightforward and easy to verify that other object detection algorithms, such as Viola–Jones method, or deep learning OD methods such as Fast-RCNN or Yolo, etc, should fit as well. Especially, the deep learning based object detection algorithm will be enough to fit a much broader range of tasks beyond Atari games. Recent researches [1] have also shown that object detection is plug-and-play modules to help improve robot perception.
>
> [1] Hu T K, Gama F, Chen T, et al. Scalable Perception-Action-Communication Loops with Convolutional and Graph Neural Networks[J]. arXiv preprint arXiv:2106.13358, 2021.
>
>
> **Q2:** The geometric operators are hand-engineered and tailored for simple 2D environments. They will be difficult to handle more complex cases like deformable objects, cooking tasks, and navigation in the 3D world.
>
>
> **Reply:**
> We respectfully yet strongly disagree with this statement. First, you show an important confusion of concepts in this statement: the key challenge of these complex scenarios is the difficulty to detect/track objects under 3D or deformation. It is *not* due to the un-applicability of the *geometric operators*. In fact, our proposed geometric operators are not tailored for 2D environments, they can be elegantly extended to more complex systems with only minor modifications.
>
> Intrinsically, all operators are defined by humans hence are "hand-engineered". Think of basic numerical operators like +,-,*,/, they are initially defined over simpler domains (rational / real numbers) then found to be applicable to complex domains (complex numbers / matrices). Their exact execution rules in the latter are slightly modified from the original one. Same case applies for our proposed geometric operators: we first define `velocity_extension` and `static_line_drawer` as the most intuitive primitives that can operate with the geometric objects. Then just as the operator "+" could be extended to matrices, these geometric operators will receive formal definitions and rigorous computations in 3D cases, while preserving their general applicabilities. Such extension simply requires 3D vector computations instead of 2D, which costs almost no additional resource.
>
> Second, CNN based state-of-the-art object detection algorithms can easily handle object detection/tracking under deformation or 3D. The objects detected by the CNN are the exact input of the proposed geometric operators. We again emphasis that, as stated in section 3, the template matching/flow-nets are *not* components of our method, but the object detector is.
>
> Furthermore, people can train a general-purpose CNN based object detector (out of many cheapy available yet provably robust benchmarks, such as Fast-RCNN/Yolo/etc.) and feed to our symbolic reasoning module *without environment-by-environment training*. However, CNN based PPO algorithms have to be trained or fine-tuned for every single environment.
>
>
>
>
>
>
>
> **Q3:** The symbolic rules do not seem to have any mechanism for long-term memory.
>
> **Reply:**
>
> We respectfully yet strongly disagree with this statement. In fact, we do include mechanisms that make our symbolic algorithm aware of the long term behavior of its entities.
>
> As already stated in the last paragraph of section 4.2, the long term memory can be captured as the attributes of the detected objects, and these attributes are maintained overtime. As an example of long term behavior, figure 11 shows in the distilled rules of Seaquest, where “player is tracking” means the player is consistently focused on the current object. Another similar case is in Fig 7 for Airstriker-Genesis.

---

### Official Review · Reviewer_VSj4 · 2021-11-02

**Correctness:** 4
**Technical Novelty And Significance:** 4
**Empirical Novelty And Significance:** 4
**Recommendation:** 8
**Confidence:** 4

**Main Review:**

The high-level procedure is relatively straightforward. Train a DRL agent directly on the environment, and then, given symbols extracted from each state, "distill" the policy into a symbolic policy (note that supervised learning applied in the context of imitation learning is known as behavioural cloning [1], and what is being done here also falls into the category of imitation learning/learning-from-demonstration). It is difficult to find symbolic policies from scratch, so this procedure makes sense, and results in policies that are (generally) more interpretable and generalise better. Without knowing the literature on symbolic regression/symbolic RL too well, to the best of my knowledge the technical contributions of this work are very good.

A weakness is the strong reliance on object detection and geometric priors for extracting/forming symbols. As it is, the algorithm cannot be applied to more complex visual domains. However, I think we should not discourage research along these directions, as the general idea of distilling black-box policies into symbolic policies has good use cases.

Some questions:
- How important is the entropy threshold? Any way for setting/tuning this automatically?
- Are there any (quantitative) ablation studies on the denoising procedure? It would be useful to know if the algorithm fails without this or just has worse performance.

Considering that the authors' method incorporates imitation learning, they may find methods such as DAgger [2] of interest. Assuming access to an expert policy that can be queried interactively, interactive imitation learning methods benefit from reducing the shift between the teacher's state-action distribution and that of the student (which is a known problem in behavioural cloning, and is potentially an issue for this method as well).

[1] Pomerleau, D. A. (1989). Alvinn: An autonomous land vehicle in a neural network. CARNEGIE-MELLON UNIV PITTSBURGH PA ARTIFICIAL INTELLIGENCE AND PSYCHOLOGY PROJECT.
[2] Ross, S., Gordon, G., & Bagnell, D. (2011, June). A reduction of imitation learning and structured prediction to no-regret online learning. In Proceedings of the fourteenth international conference on artificial intelligence and statistics (pp. 627-635). JMLR Workshop and Conference Proceedings.

**Summary Of The Paper:**

The authors propose a novel method for finding symbolic policies for image-based RL environments. The method consists of training a traditional DRL algorithm, and then "distilling" this policy by searching for a symbolic policy that mimics the DRL teacher agent. This can result not only in interpretable, lightweight policies, but also policies that generalise better than the original teacher policy.

**Summary Of The Review:**

I believe the authors tackle an interesting and relevant problem in RL, combining both deep learning and symbolic approaches to good effect. To the best of my knowledge, there are solid contributions in this work, and I would recommend accepting this paper.

---

> ### Author Response · Authors · 2021-11-18
> **Authors' response to reviewer VSj4**
>
> Dear reviewer VSj4:
>
> We appreciate your acknowledgement of this work! We hope our responses below help to further clarify the proposed methodology.
>
> **Q1:** What is the importance of the entropy threshold, and is there any way for setting/tuning this automatically?
>
> **Reply:**
>
> The role of the entropy threshold is to judge when to assign an action node or a condition node. A higher threshold leads to a looser condition for assigning an action node, and leads to shallower, less distinctive, and more “smoothed” policy tree. It is analogous to the hyperparameter that basically any machine learning model would require. In our experiments, we set it as the best performing one with 10 fold grid search from 0 to 1 for different environments.
>
> Potential automatic ways for tuning this hyperparameter include using the Hyperparameter Optimization (HPO) approaches. For example, one can distill a list of experiments with different thresholds, then infer the optimal value for this experiment using Bayesian optimization algorithm (BOA), Gaussian Process Regression (GPR), or training an LSTM to predict the best possible threshold.
>
>
>
> **Q2:** It is recommended to provide some quantitative ablation studies on the denoising procedure.
>
> **Reply:**
> This is a good suggestion. We have made a new series of experiments to verify the contribution of the denoising procedure. Specifically, we take CircusCharlie and Pong as the testing environments, and set the denoise probability to be zero, and compare the difference of the distilled symbolic policy. The comparison for the average leaf node depth and the performance of the resulting policy are put in the table below. We empirically observed that the leaf nodes are deeper without denoise, but the performance did not significantly drop. We also found that the deeper leaf nodes in most cases correspond to states that do not affect the reward much. For example, in Pong, the tree grows more in the cases when the ball is on the other half of the court, while when the ball is near the agent’s side, the agent takes almost deterministic actions and the tree effectively stays the same. We expect the performance may drop more on some other environments if without denoising, dependent on the stability of the CNN teacher in that specific environment.
>
>
> |  | avg depth (denoise prop 0.2) | reward (denoise prop 0.2)  | avg depth (denoise prop 0.0) | reward (denoise prop 0.0) |
> |---|---|---|---|---|
> | CircusCharlie | 4.28  | 7610  |  5.53  |  7210  |
> | Pong  | 3.25  |  16.50 |  5.61  | 15.00  |

---

### Official Review · Reviewer_cXsw · 2021-11-02

**Correctness:** 3
**Technical Novelty And Significance:** 4
**Empirical Novelty And Significance:** 4
**Recommendation:** 6
**Confidence:** 4

**Main Review:**

**Review update**
After reading through the authors' comments, I've decided to raise my score to a 6. I am willing to support acceptance of this paper. I believe it offers an original and definite contribution to an important but underdeveloped problem. It by no means solves all the problems need to bridge symbolic and black box deep RL, but having this work out there will allow others to build upon it.

=========================

This was an interesting paper that shows the potential of symbolic RL, both in yielding better generalisation and in offering more human-interpretable policies. The authors' approach of distilling a black-box teacher policy into a symbolic one is quite interesting. The approach by no means bridges the gap entirely between black-box and symbolic RL, and the approach add some complexity by introducing extra hyperparameters including the maximum depth of the decision tree, an entropy threshold, and the set of symbolic building blocks that the symbolic policy can work with. The main strengths and weaknesses of the paper are as follows.

**Strengths:**
* original approach to distill black-box policies into symbolic ones
* the symbolic policies found are human-interpretable
* the symbolic policies are able to de-noise teacher policies, leading to qualitatively better policies (e.g. the symbolic agents take fewer harmless but wasteful actions at each time step; this means that we see, e.g., less paddle jitter in Pong when ball is far away, which we often see in PPO and other deep RL agents instead.) Perhaps the de-noising can help induce more efficient gaits in the Mujoco tasks.
* the symbolic policies show superior transfer compared to their original teacher policies, showing the promise of symbolic policies in offering better generalisation

**Weaknesses:**
* some of the empirical results could be cleaner. This is the main weakness of the paper and is what makes me hesitate in fully recommending the paper for acceptance at this stage. Some of these issues, I believe, can be remedied by the authors during the discussion phase.
    * how many random seeds were used for PPO, A2C and the distillation for all tasks? I see variances for PPO and A2C but not the distillation (which does use stochastic steps in e.g. Algo 1). It'd be helpful to know how robust is symbolic policy's superior performance to the choice of random seeds.
    * the Atari results (Pong and Seaquest) are non-standard, making comparison with baselines in the literature difficult. First of all, they only allowed the agent a single life, and they train for 10M steps only (whereas 200M is more standard for published Atari results). Because of this, the paper's scores for PPO and A2C are far lower than what is in the literature.
    * moreover, why is PPO’s learning curve for Pong far slower than that in Schulman et al, 2017 (https://arxiv.org/pdf/1707.06347.pdf). They seem to get near the max score of 20 points within 10M time steps.
    * it would also be helpful to know whether the authors are using an open source release of PPO and A2C or a reimplementation. If it is an open source release, could the authors reference which implementations they used?
* the authors gave a nice demonstration of the superior generalisation of their symbolic policies. The authors could also look to the Procgen Suite of games for further tasks to demonstrate generalisation. These were designed explicitly to train agents on one set of levels and test on held-out levels to evaluate agents' ability to generalise.
* the authors mention tasks with continuous action spaces but offer no results on these (all reported results were on video games, which have discrete action spaces). What is the reason for this? If their approach was not as successful on continuous action tasks, it'd be helpful to report this and offer possible explanations why in the Discussion.
* it'd be helpful to have some metric of the training complexity needed to distill a symbolic policy.
* there were steps in the distillation algorithm that were unclear:
    * what drives improvement in each iteration of Step 2 (Guess & Observe)? Particularly, what drives improvement of the geometric relations?
    * how are candidate conditions for policy’s condition nodes chosen? It would seem like the list of possible conditions to pick is infinite, making search difficult.
* the authors reference Botvinick et al (2009), but this reference seems completely irrelevant to the authors' discussion. The Botvinick et al paper is about hierarchical RL (hRL) in neuroscience. First of all, hRL does not seem relevant to what the authors are doing here -- hRL concerns hierarchical structure in policies where a higher level policy is defined in terms of macro actions, each of which defines a sub-policy in terms of primitive actions. This seems irrelevant to the authors' work here. Secondly, it is unclear why the authors chose to cite this particular hRL papers and not others, including Sutton, Precup, Singh (1999) on the options framework, Dietterich (1999) on MAXQ, Dayan & Hinton (1992) on feudal RL, or any of the many papers on deep hRL that have been published since.
* minor point: the writing, while still highly readable, could benefit stylistically from the help of a proofreader more fluent in English. There were small grammatical errors (that did not hinder comprehension or ease of reading), but there were also awkward word choices. Here are just a few examples:
    * p5, line 1 - “the learning target 1 rules the [geometric symbols→numerical state] step”: perhaps say "1 determines the learning target for the [geometric symbols→numerical state] step". Likewise for "targets 2 and 3 rules the ..." on the line below.
    * p5, 2nd paragraph of the subsection "The geometric operator search space" - "the nearest class-$i$ ($i$ is given) object from the **protagonist** in the current observation": usually, "protagonist" refers to the main character of a story. Perhaps reword this to "from **a reference object** in the current observation"
    * p6, last paragraph - "father nodes": the more common terminology is "ancestral nodes".


**Summary Of The Paper:**

This paper proposes to bridge the gap between black box deep reinforcement learning (RL) and more interpretable symbolic RL by distilling learned deep RL policies into a symbolic decision tree. While the symbolic entities for each task are predefined, their algorithm is able to extract out a symbolic representation of the visual input as well as a symbolic policy in terms of a decision tree. The authors show that the distilled policy is highly human interpretable and even generalises better than the original teacher policy to tasks that are visually different but symbolically identical to the original training task.

**Summary Of The Review:**

I believe this paper offers sufficient contributions to the field to potentially be accepted. Although it by no means solves all problems in this domain, it does help toward bridging the gap between black-box deep RL policies and more interpretable, symbolic RL policies. For me, the paper's primary contributions are the use of an original distillation procedure to translate black-box policies into symbolic ones, which offer the benefit of being more human interpretable and generalising better to tasks outside the training set. So this paper does have promise. But what prevents me from fully supporting it just yet are issues with the empirical evaluation. I would, in particular, like to ensure that the PPO and A2C baselines are accurately generated and that the results are robust to the choice of random seeds before I can fully support the paper's acceptance.

---

> ### Author Response · Authors · 2021-11-18
> **Authors' response to reviewer cXsw (part 2)**
>
> Dear reviewer cXsw:
>
> Due to length limitation, we continue our response session here.
>
>
> **Q5:** Can a metric be defined to quantify the training complexity needed to distill a symbolic policy?
>
> **Reply:**
> We appreciate this suggestion. We define the training complexity as the following:
>
> Training complexity = (the length of the finally recovered equations) * (the number of finally recovered operator types) / (the number of all operator types) = (tree depth) * (tree size) / (the number of all operator types).
>
> The number of operator types is 30 (19 geometric operators and 11 numerical operators), the number of finally recovered operator types (tree size) and the length of the finally recovered equations (tree depth), and the corresponding estimated training complexity is provided in the following table. Intuitively, this metric means longer the equation the more ratio of desired operator types, the more complex the training procedure is.
>
> | Environment          | Tree Depth | Tree Size | Training Complexity |
> |----------------------|------------|-----------|---------------------|
> | CircusCharlie-Nes    | 4          | 6         | 0.8                 |
> | Airstriker-Genesis   | 4          | 7         | 0.93                |
> | Pong-Atari2600       | 3          | 3         | 0.3                 |
> | AdventureIsland3-Nes | 6          | 8         | 1.6                 |
> | AstroRoboSasa-Nes    | 3          | 5         | 0.5                 |
> | Seaquest-Atari2600   | 6          | 9         | 1.8                 |
>
>
> **Q6:** What drives improvement in each iteration of Step 2 - “Guess and Observe”?
>
> **Reply:** Step 2 “Guess and Observe” does not have any type of learning in it. After randomly sampling from the possible geometric and numerical functions to create the numerical state and performing Step 3 “Solve”, if the resultant symbolic policy does not match the teacher agent’s behaviour, we directly discard the numerical state and start over Step 2 again. We have added a new *appendix A*, which includes more justifications of the motivations and methodologies, please kindly check it out.
>
>
> **Q7:** How are candidate conditions for policy’s condition nodes chosen in infinite space?
>
> **Reply:**
> In fact, the selection space is not infinitely large. Precisely, we used 11 types of geometric operators and 19 types of numerical operators. The number of objects in each frame is usually less than 10, let’s assume there are maximally 20 objects. At each decision node, we constrain the number of operators and the objects (operands) to no more than 3. This leads to 20^3*30^3 = 2.16e8 maximum choices at each node. In practice, not all operands/operators are compatible, and the objects are usually less than the above number. With elimination of incompatible combinations, the search space is smaller.
>
>
>
> **Q8:** The author’s reference Botvinick et al (2009) seems irrelevant to the authors' discussion.
>
> **Reply:**
> We agree with you that the method in Botvinick et al (2009) is different from the proposed one. We indeed didn't mention this paper did a relevant job, only the proposed multi-level sub-policy extended itself in a hierarchical shape, similar to our distilled policy. We have additionally cited Precup, Singh (1999) and Dietterich (1999) as references.
>
>
>
> **Q9:** Minor points on the writing
> **Reply:** Thank you for pointing these out. We have now addressed these issues in the updated PDF.

---

> > ### Comment · Reviewer_cXsw · 2021-11-23
> > **Thank you for your reply**
> >
> > I'd like to thank the authors for their thorough reply to my comments. There is one point I'd like to follow up on, and this concerns the absence of learning in the "Guess and Observe" phase of the algorithm. As the authors point out, the geometric search space is prohibitively large, so randomly guessing the correct geometric operators (without any learning or optimisation) would seem like finding a needle in an enormous haystack. So while I'm not disputing any of the authors' empirical results, I am surprised from a theoretical point of view that this is able to work. I was wondering if the authors could provide more insight on this paradox. Is it the case that there is a sizeable number of alternative geometric relations that can work for the same frames, and different instantiations of random searches on this data set will yield different geometric relations? Or is it the case that the symbolic policy must find a specific set of geometric relations to work? From the authors' 10x repetitions of symbolic distillation, what do the authors find?

---

> > > ### Author Response · Authors · 2021-11-24
> > > **Addressing follow up questions**
> > >
> > > Dear reviewer cXsw:
> > >
> > > We appreciate your response a lot!
> > >
> > > We hold different views in the statement "there is an absence of learning in the symbolic distillation algorithm". The symbolic distillation is gradient free, but gradient free is not equivalent to "*learningless*". The gradient free is due to the nature of the symbolic (the compositions of *discrete* operators). It is a discrete optimization problem generally believed to be NP-hard [1]. In addition, the proposed approach belongs to the Genetic Programming (GP) type algorithm as it evolves and mutates the candidates, similar to [2][3].
> > >
> > >
> > > [1] Mundhenk T N, Landajuela M, Glatt R, et al. Symbolic Regression via Deep Reinforcement Learning Enhanced Genetic Programming Seeding[C]//Thirty-Fifth Conference on Neural Information Processing Systems. 2021.
> > >
> > > [2] Virgolin M, Alderliesten T, Witteveen C, et al. Improving model-based genetic programming for symbolic regression of small expressions[J]. Evolutionary computation, 2021, 29(2): 211-237.
> > >
> > > [3] Cranmer M, Sanchez-Gonzalez A, Battaglia P, et al. Discovering symbolic models from deep learning with inductive biases[J]. arXiv preprint arXiv:2006.11287, 2020.
> > >
> > >
> > > For your question "*Is there a sizeable number of alternative geometric relations that can work, or does the symbolic policy must find a specific set of geometric relations to work*", from the 10x repetitions, we found two types of variants other than the reported results: take the example of:
> > >
> > >  $(x>50)*a\_1 + \neg(x>50)*a\_{default}$
> > >
> > > The first one is when the $(x>50)$ are also found as $(-x<-50)$. The second one is when the subtree of $a\_{default}$ are further divided into  $(y>0)*a\_2 + \neg(y>0)*a\_{default}$, but in the environment, either take $a\_2$ or $a\_{default}$ leads to the same reward (no substantial difference from the simulator point of view). Our findings shows that there is a sizeable set that leads to valid results, up to the equivalence of math or the simulations.
> > >
> > >
> > > We are glad to further address whatever questions you might have.
> > >
> > > Best,
> > >
> > > Authors of paper942

---

> > > > ### Comment · Reviewer_cXsw · 2021-11-24
> > > > **Further follow up**
> > > >
> > > > Thank you again for your reply. But I'm still a bit confused by this and wanted to verify things. Because in your initial reply to my Q6 (see above), you said that: *Step 2 “Guess and Observe” does not have any type of learning in it. After randomly sampling from the possible geometric and numerical functions to create the numerical state and performing Step 3 “Solve”, if the resultant symbolic policy does not match the teacher agent’s behaviour, we directly discard the numerical state and start over Step 2 again.*
> > > >
> > > > But now you are saying that "Guess and Observe" does perform learning (albeit gradient free) -- so I am confused. Perhaps you mistook my question for the "Solve" phase of your algorithm (Table 1), which I do appreciate as being a genetic programming algorithm. Could you please clarify? And if "Guess and Observe" is also performing a form of genetic programming, could you explain how, as it wasn't clear from the manuscript? In particular, when you have to make a new "guess" are you simply throwing out your initial guess of the geometric operators and starting from scratch, or are you somehow mutating the "fittest" operators from the previous generation?

---

> > > > > ### Author Response · Authors · 2021-11-24
> > > > > **Further responses**
> > > > >
> > > > > Dear reviewer cXsw:
> > > > >
> > > > > Thank you for this question.  Let us clarify them as follows:
> > > > >
> > > > > 1. You were right that the "*Solve*" stage is the core algorithm of Genetic programming, and this stage contains the gradient free learning;
> > > > >
> > > > > 2. The "*Guess and Observe*" does not include random mutation of the operators, hence it does not learn the rule by itself directly. However, what happens in the "*Guess and Observe*" stage is that a combinations of the geometric relationships to be measured are chosen. If certain critical geometric relationship is missed in this step, the following step will fail and the overall RoundTourMix algorithm will return to the "*Guess and Observe*" step again. Therefore, the "*Guess and Observe*" could be treated as an "outer loop" of genetic programming, hence it contains learning when integrated into the entire algorithm.
> > > > >
> > > > > As a concrete example, in the Pong, the "*Guess and Observe*" could randomly choose to  observe the x/y-coordinate of the pong object, the velocity of the pong object, and/or other geometric features. In the procedure, it is possible that the velocity is missed (not measured), hence the rules learned by the following steps will be suboptimal. In the next episode of RoundTourMix, the correct geometric relations could be all included, then a better rule is learned.
> > > > >
> > > > > We always do neighborhood sampling of operator combinations (add/drop subtree), e.g., in line 13/25 of the algorithm table. The motivation behind neighborhood sampling is that general GP algorithms do minor mutations under large discrete space.
> > > > >
> > > > >
> > > > >
> > > > > Best,
> > > > >
> > > > > Authors of paper942

---

> > > > > > ### Comment · Reviewer_cXsw · 2021-11-25
> > > > > > **Thank you for the clarification**
> > > > > >
> > > > > > Thank you for the clarification. My concern is that the more geometric relations you need to include, the harder it will be to guess them all if you're relying on chance alone. For instance, suppose you needed to observe features {A, B, C, D, E} (to be concrete, A could be ball position, B could be ball velocity, C could be paddle position, etc). In one guess, you might get {A and D} right but miss {B, C, and E}. In the second attempt, you may get {A, B, and E}, but now you've lost D, etc. And it may take quite a few guesses before you finally get the combination {A,B,C,D,E} because the probability of guessing that combination is low. And if the probability of getting the right set of geometric relations is low, then the outer loop of your algorithm would be slow, and this would be a limitation of your algorithm. As a concrete analogy, it's a bit like drawing four cards from a deck of cards repeatedly until you get four queens -- it will happen, but it may take a lot of iterations before it does. So there's a question of whether "Guess and Observe" can scale to scenarios where the probability of including the right features is small.

---

> > > > > > > ### Author Response · Authors · 2021-11-25
> > > > > > > **Further explanations**
> > > > > > >
> > > > > > > Dear reviewer cXsw:
> > > > > > >
> > > > > > > Thank you for your feedback. The scaling is not a significant issue, as the number of algorithm output is polynomial w.r.t. the number of objects, and the number of objects is usually small.
> > > > > > >
> > > > > > > Denote the set of operands that are actually picked and used in the guess and observe stage as $\mathcal{U}$, and the number of all valid geometric operands to consider as $\mathcal{V}$, $|\mathcal{V}|=v$. For simplicity, consider all the decision node take three operands, one math operator and one boolean operator (e.g., $a+b<c$). Denote the number of math operator and boolean operators are $m$ and $b$ respectfully. The number of combination is
> > > > > > > $\frac{1}{2}v^3mb$, where the $\frac{1}{2}$ means the commutation duplicate removal. On average, we could assume the number of ground truth is $1$ among these combinations. In order to setup metric for the learning complexity at this node, At certain branching node of the solve stage, we compute the number of all possible algorithm output. Assume the depth of this node is $d$ and denote the number of all possible subtrees yielded by the algorithm is $N_d$.
> > > > > > >
> > > > > > > To compute $N_d$, there are two cases: first, if the branching is incorrect, the teacher's action under the branching condition will be nearly random, which will on average lead to a single child node option (default action). Second, if the branching is correct, the tree will on average be branched further, which leads to $N_{d+1}$ child node options. The number of branching choices that leads to these two cases are approximately  $\frac{1}{2}v^3mb-2$ and $2$. Therefore, we have:
> > > > > > >
> > > > > > > $N_d=(\frac{1}{2}v^3mb-1)\cdot1+2\cdot N_{d+1}$
> > > > > > >
> > > > > > > Denote the maximum depth as $D$, then $N_D=1$. Further denote $K=\frac{1}{2}v^3mb-1$.
> > > > > > >
> > > > > > > Next we consider the complexity of the outer loop, and the influence of not picking the correct subset during the Guess and Observe stage. If the correct geometric relationships are picked, $\mathcal{U} \subseteq \mathcal{V}$, then
> > > > > > > $N_d=K\cdot1+2\cdot N_{d+1}$ can be satisfied in every iteration. Solving the above iterative relation, we get the average number of output at the root node of the algorithm is:
> > > > > > >
> > > > > > > $N_0=(1+K)2^{D-1}-K$
> > > > > > >
> > > > > > > In other cases, if not all essential relationships are used by the Guess and Observe, $\mathcal{U} \nsubseteq \mathcal{V}$, then the tree will early stop at depth $\bar{D}=\lfloor \frac{|\mathcal{V}\cap\mathcal{U}| }{|\mathcal{V}|}D \rfloor$, since the missing of essential operands will happen at shallower depths of search, leading to shallow trees. For simplicity, we assume the missing of operands happens at the first layer ($\bar{D}=1$, $N_0=1$), or the second layer ($\bar{D}=2$, $N_0=K+2$), which lead to an average case of $N_0=3/2+K/2$.
> > > > > > >
> > > > > > > Unify the two cases, we can see that the overall possible output combinations are:
> > > > > > >
> > > > > > > $N_{output} = p(\mathcal{U} \subseteq \mathcal{V})\left((1+K)2^{D-1}-K\right) +p(\mathcal{U} \nsubseteq \mathcal{V})\left( 3/2+K/2 \right)$
> > > > > > >
> > > > > > > which grows in cubic to the number of operands. In practice, the maximum depth $D$ is only moderately deep (with Seaquest and AdventureIsland3 being 6, the deepest across our selected environments), and the number of objects in the frame is usually less than 10.

---

> > > > > > > > ### Comment · Reviewer_cXsw · 2021-11-26
> > > > > > > > **Not sure if I fully agree with the argument above, but will support acceptance of paper**
> > > > > > > >
> > > > > > > > Dear authors,
> > > > > > > >
> > > > > > > > Thank you again for the additional comments. I'm not sure it fully addresses the issue I brought up, mainly because it hinges on $p(\mathcal{U} \subseteq \mathcal{V})$. The outer loop will not terminate until this event is true, and on average, that happens once every $1/p(\mathcal{U} \subseteq \mathcal{V})$ guesses which means you'd need as many outer loop iterations before you guess the correct set of operators and operands.
> > > > > > > >
> > > > > > > > That issue aside, I've still decided to support accepting the paper -- I believe it offers a definite and original contribution to an important but underdeveloped problem. I think there'd be an interest in the community in seeing this work and building upon it, and I've raised my score accordingly.

---

> > > > > > > > > ### Author Response · Authors · 2021-11-26
> > > > > > > > > **Thank you**
> > > > > > > > >
> > > > > > > > > Comment: Dear reviewer cXsw:
> > > > > > > > >
> > > > > > > > > Thank you for the precious endorsement and review!
> > > > > > > > >
> > > > > > > > > Best,
> > > > > > > > >
> > > > > > > > > Authors of paper 942

---

> ### Author Response · Authors · 2021-11-18
> **Authors' response to reviewer cXsw (part 1)**
>
> Dear reviewer cXsw:
>
> We sincerely appreciate your feedback, and we have answered your questions in detail below. We hope our responses help to address your concerns.
>
> **Q1:** How many random seeds were used? How robust is symbolic policy's superior performance to the choice of random seeds?
> There is variances for PPO and A2C but not the distillation
>
> **Reply:** There are two sources of randomness. The one from training the CNN (teacher) agent, and the one from the distillation of symbolic policy. In our experiments we assume that traditional RL training leads to stable CNN policies, hence we train all the CNN agents only once. The shadow visible in figure 4 is the real time reward while the solid line is the running mean.
>
> To verify the case for symbolic distillation procedure, we take the CircusCharlie as a testbench,  and run the symbolic distillation for 10 times. For each run, we used a different training dataset (generated from the behavior of the teacher with different random seeds for the environment). The resulting skeleton of the distilled rule is the same, with only mathematically equivalent variants of equations (i.e., $x<y$ and $-x>-y$).
>
>
> **Q2:** The Atari results given in the paper are non-standard:
>
> **(a)** The authors only allowed the agent to have a single life
>
> **Reply:**
> As a pioneering work, we focused on the research problem of developing the symbolic distillation algorithm for visual RL tasks. The choice of single life still offers fair comparison between the PPO, A2C and the distilled symbolic policy, since the observation is the same for them. On the other hand, such engineering choice is more implementation-friendly to the object detection head needed for the symbolic rule (it avoids later stage frame flickering).
>
>
> We have offered two benchmarks (A2C and PPO) to mitigate the non-standard training settings, which are all set to be standard configurations [rl-baselines3-zoo by stable-baselines3](https://github.com/DLR-RM/rl-baselines3-zoo). The training curves for these two standards RL methods offer a benchmark under our settings. Additionally, we have provided the videos of the behavior of the trained PPO agent (in the initially submitted supplementary materials) to visualize its verifiable performance.
>
>
>
>
> **(b)** They train for 10M steps only
>
> **Reply:** We use the default hyperparameters as provided by [rl-baselines3-zoo by stable-baselines3](https://github.com/DLR-RM/rl-baselines3-zoo), which uses 10M training steps for these environments.
>
> **(c)** The PPO agent’s learning curve for Pong far slower than Schulman et al, 2017
>
> **Reply:** We used Retro-Gym as our environment platform instead of Atari-Gym. The deviation from the commonly-trained PPO/A2C agents is expected. This is due to Retro-Gym environments using the much harder multi-discrete action space over just discrete as present in Atari-Gym. Another possible reason could be due to some visual differences between the same games from the two backends. The third minor possibility is the differences between the Retro ALE backend interface rather than the usual Atari-Py ALE backends.
>
> **(d)** Have the authors used an open source implementation for PPO and A2C?
>
> **Reply:** Yes, we follow the implementation of [rl-baselines3-zoo by stable-baselines3](https://github.com/DLR-RM/rl-baselines3-zoo). We follow the default hyperparameters provided by their implementation: https://github.com/DLR-RM/rl-baselines3-zoo/blob/64d4625599d3244308f643810fd57d240aeeac58/hyperparams/ppo.yml#L10
>
> **Q3:** The Procgen suite of games might be of interest to the authors.
>
> **Reply:** We genuinely thank you for your recommendation. We believe Procgen will help to better evaluate the generalization ability, and are glad to adopt it in our future works.
>
> **Q4:** The paper mentions continuous action space but didn’t compare it.
>
> **Reply:** We didn’t compare the continuous action space because the simulation platform (Retro) does not contain continuous action space environments throughout the 1000+ games. For all the Retro-Gym environments, the standard API offers discrete action space. We mentioned in the paper that we take the Cross-Entropy loss as the metric under the discrete action space, and also note the choice to use Mean Squared Error loss given the potentially continuous action space in other environments. Thanks for pointing this out and we’ve improved the preciseness of the statement.

---

### Official Review · Reviewer_wezQ · 2021-11-03

**Correctness:** 2
**Technical Novelty And Significance:** 2
**Empirical Novelty And Significance:** 2
**Recommendation:** 3
**Confidence:** 4

**Main Review:**

The paper can be hard to follow at times, and could be simplified and clarified (e.g. 4.1 could be a figure, 4.2 could benefit from adding one). Another example is that Step 2 "Guess and Observe" of RoundTourMix is not defined in enough details (how many geometric features? picked how?) for the results to be reproduced.

There is a bit of missing related work. Specifically, "State of the Art Control of Atari Games Using Shallow Reinforcement Learning" (Liang et al. 2015) that trains linear models on higher level (almost symbolic) features extracted similarly as in this paper. In fact, those features are from the origin of Atari Learning Environment. A similar work is "Planning From Pixels in Atari With Learned Symbolic Representations" (Dittadi et al. 2020). Additionally, the litterature of inductive logic programming and case-based reasoning applied to (video) games is relevant, as inferring case-based policies from demonstrations has been an active area of research (e.g. "Case-Based Planning and Execution for Real-Time Strategy Games" (Ontanon et al. 2007)).

The experimental results show that a more robust policy than the PPO and A2C learned policies gets extracted for most of the subset of games that the authors picked (also exemplified in Figure 4.b.). But, this does not necessarily justify the choices of the RoundTourMix algorithm (central contribution of the article). Indeed there is no justification why one has to make a random selection (2. Guess and Observe) of the geometric relations, and why we can't simply be exhaustive for those games (I believe we can at least for Pong and Seaquest). This is my biggest criticism of this paper: there is very little justification and/or empirical validation for the many choices of the RoundTourMix heuristic.

"Transferability validation" is an interesting experiment, that shows that the symbolic policy (distilled from a CNN trained with PPO on AdventureIsland 3) transfers to a similar but different game, AdventureIsland 2. Sadly, there was no apples to apples comparison by having a PPO trained policy simply plugged onto the geometric symbol representation. This would have allowed to show how much of the transferability comes from the features and how much comes from the "if-this-then-that" policy.

**Summary Of The Paper:**

The paper presents a method for distillating potentially any policy into a symbolic policy (in prepositional logic + basic maths operations), applied in 2D video games based on the Atari, NES, and Genesis emulators. The procedure (RoundTourMix) consists in applying random search to iteratively i) select the features of interests, and ii) solve for a symbolic policy (finding conditions / FSM induction) that works with those features. The teacher policy in all experiments in a neural network trained with PPO. The conversion from pixels to objects ("geometric features") is assumed solved. Experimental results are presented on 7 games.

**Summary Of The Review:**

The central contribution of the paper (the RoundTourMix distillation procedure) seems ad-hoc and many of its decisions are not well-enough justified. The experimental validation has some flaws that need to be addressed to make it a convincing contribution.

Finally, this is honestly not a great fit (topically) for ICLR and would be better suited in a games (e.g. AAAI AIIDE, IEEE CoG) conference or workshop. In its current form, the contribution is too "heuristic-y" for publication at ICLR.

---

> ### Author Response · Authors · 2021-11-18
> **Authors' response to reviewer wezQ**
>
> Dear reviewer wezQ:
>
> We genuinely appreciate your suggestions to strengthen our paper. We have tried our utmost to address all raised questions, and have also amended our Appendix in the updated PDF to include more information.
>
> **Q1:** There is no justification for random selections in Step 2 “Guess and Observe. Why does the proposed method rather not use exhaustive search?
>
> **Reply:** Indeed, doing exhaustive search will guarantee the emergence of good performing policies upon unlimited hardware and software resources. However as we have pointed out at the bottom of page 5 - “One possible difficulty of the symbolic distillation procedure is the huge size …”, the search space of the geometric representation is exponentially large. Therefore random selection is the only feasible solution compared to exhaustive search. In addition, the random search paradigm empirically provides reliable results compared to exhaustive searching according to our experiments.
>
> Precisely, we have a total of 11 different operators in the geometric operator search space and 19 different operators in the numerical operator search space. If we compose symbolic compositions of length 20, the total possible policies are (11+19)^20 = 3.48e29.
>
> Moreover, any exhaustive or brute force search goes against the spirit of reinforcement learning as there is no “learning” involved. Traditional reinforcement learning does not employ any such exhaustive methods due to the same reason of the large number of possible policies.
>
> **Q2:** It would be appreciable to have results on a PPO trained policy simply plugged onto the geometric symbol representation.
>
> **Reply:**
>
> We are actively working on it and we hope to produce the results soon. We will update the comparison once a thorough comparison is complete.
>
> **Q3:** Step 2 "Guess and Observe" of RoundTourMix is not defined in enough detail. Also how many geometric features are part of the search space?
>
> **Reply:** We iterate over the image from the teacher's behaviour, evaluate the randomly sampled geometric and numerical relations. For each frame we create a vector of size 16 of possible logical operands which are then later on used as the numerical state for symbolic distillation. This is quite sufficient from our empirical results as the highest number of operands distilled for any successful game does not cross 11. We have also included a section in the appendix for further information.
>
> We have a total of 11 different functions in the geometric operator search space and 19 different functions in the numerical operator search space. We provide the justification of choosing each of these operators along with two examples in Section 4.2 (page 5).
>
> **Q4:** The paper can be hard to follow at times, and could be simplified and clarified. Also there is very little justification and/or empirical validation for the many choices of the RoundTourMix.
>
> **Reply:** We have added detailed explanations in appendix A.2 of the updated PDF, titled **Motivations behind the Algorithmic Choices**. The newly added justifications include: *Justification for Random Selection. Guess and Observe. Explanation of Branching Conditions*. We hope these justifications solve your concerns, please kindly have a check. On the other hand, the overall presentation of this paper is acknowledged by reviewer AL2N, reviewer GGAU, and reviewer VSj4 as well organized: “this paper is easy to follow, clearly presented, and relatively straightforward”.
>
> **Q5:** There is a bit of missing related work.
>
> **Reply:** We thank the reviewer for recommending these relevant works. After careful check, these works do not degrade the novelty and contributions of our work. We have added citations and have compared them as follows and in the updated PDF.
>
> [1] offers a high level rule over NN outputs, but it is not by distilling the rule of a learned RL model. Moreover, [1] only uses simple linear representations whereas our method produces non-linear symbolic rules depicting richer expressiveness.
> [2] uses a VAE to extract relevant pixel features from screen states which are then provided to the planning algorithm. Firstly, this approach still remains as a black-box form while our algorithm can come up with interpretable and robust white-box rules. Secondly there is no aspect of memory or temporal-dependent behaviour which is crucial for many 2D games.
> [3] follows a case-based expert-behaviour retrieval approach. Our method on the other hand requires nothing more than a cheap object detector which can locate the different entities of the game. The complete symbolic tree can be distilled out directly from the state-action pairs of the offline teacher dataset.
>
> [1] State of the Art Control of Atari Games Using Shallow Reinforcement Learning (Liang et al. 2015)
> [2] Planning From Pixels in Atari With Learned Symbolic Representations (Dittadi et al. 2020).
> [3] Case-Based Planning and Execution for Real-Time Strategy Games (Ontanon et al. 2007)).

---

> ### Author Response · Authors · 2021-11-27
> **Sincerely expecting further discussions**
>
> Dear Reviewer wezQ,
>
> We would like to thank you for the constructive comments in your review. As a follow-up on our responses, we kindly remind you that the discussion period is ending soon. We hope to use this open response period to discuss the paper to solve the concerns and enhance the quality of our work. Have you gotten a chance to read our responses below, which attempt to address all of your issues?
>
> In response to your suggestion to compare and contrast  previous studies on symbolic and interpretable learning in an RL setting, we have added Blob-PROST, VAE-IW, and Case-Based Retrieval in both the response and our updated PDF. We have answered your questions regarding why an exhaustive search is prohibitive and included a section in our main text regarding the complexity analysis of the learning process.
>
> In response to Q2, we now show the results of the geometric symbol representation plugged as inputs to a PPO algorithm (termed Symbols-PPO).
>
> | Environment | CNN-PPO | Symbols-PPO | Distilled CNN-PPO (Ours) |
> |---|---|---|---|
> | Pong-Atari2600 | -9.3 | -8.8 | 17 |
> | Seaquest-Atari2600 | 220.4 | 184.4 | 940 |
>
> Our results convey that a CNN-based PPO and a geometric symbols-based PPO have comparable performances. It also clearly shows the superiority of our approach, which accurately distills out symbolic rules while selectively rejecting the noisy teacher behaviors, given just an offline dataset of symbols and a set of geometric operators.
>
> We have also added detailed explanations on the justification and motivations behind our algorithmic choices in our appendix and have addressed all typos you mentioned. Please kindly have a check on the updated PDF.
>
> We sincerely hope to have further discussions with you to see if our response solves the concerns. We would be more than happy to provide more information or clarification, should it be necessary, and hope our paper could receive a positive and fair assessment.
>
> Best,
> Authors of paper 942

---

> ### Author Response · Authors · 2021-11-29
> **Sincerely expecting further discussions with reviewer wezQ**
>
> Dear Reviewer wezQ,
>
> We kindly noticed that your main concern for this paper is "decisions are not well-enough justified". In response to this, we have answered them in detail as above, as well as in the updated PDF. As the discussing phase is ending soon, could you please kindly have a look, and let us know if we have addressed your concerns?
>
> Looking forward to your reply!
>
> Best,
>
> Authors of paper 942

---

### Decision · Program_Chairs · 2022-01-20

**Decision:**

Reject

**Comment:**

After going over the reviews and the rebuttal, and skimming the paper, I feel like unfortunately this paper is not ready to be accepted.

My reasoning is as follows. I feel the comparison with A2C and PPO is not and should not be the main target of the work. Of course they are good to have as reference points, and they should be in the paper. But the work is not trying to claim that the distilled symbolic policy is more data efficient (or outperforms these methods). If that would be the point, that one has questions similar to reviewer cXsw about these baselines maybe underperforming (compared to other published work). Maybe this is due to a change in setup as argued by the rebuttal, nevertheless this makes comparison and understanding the results difficult. The other argument is that  A2C / PPO are not the most efficient DRL methods for atari.  Lastly, is the question of distilled symbolic policy having access to an expert, making this not an apples to apples comparison.
But as I said, and I think this is the point of the authors as well, this is not the point of the paper. But then I find the results not being sufficiently contextualized either by comparing to other methods in this space, or various ablation studies to motivate the choices taken by the authors. Similar points were raised by other reviewers (wezQ, cXsw). Some of these ablations have been brought forward in the rebuttal, but I think they should be a more central part of the work and implies considerable edits to the paper.
I think the stance that the object identification is decoupled from the symbolic policy is also a bit dangerous. I.e. a learned object identifier (particularly in a visual more complex setting) will have different failure modes, which will affect the policy. I think having a paragraph discussing the issues raised by reviewer AL2N would actually strengthen the paper, and being open about open questions/weaknesses. Alternatively additional ablation or experiments in either other kind of environments (e.g. 3D or environments with occlusion) or just assuming some form of failure at segmentation the visual stream into objects to show robustness would be of interest.

Overall I urge the authors to resubmit their work after properly integrating some of the feedback. In particular focusing on ablation studies or having baselines that are more similar in spirit or at least being more explicit of how it compares with existing work and what aspect of that existing work is trying to fix. For e.g. part of the approach is that it relies on distillation rather than dealing with the RL objective (as other methods might try to do). Now if you take those methods, but you phrase them in a distillation process how would they do? I don’t know if all of this needs to be done, but it just feel as a work to be less grounded and sufficiently far to other existing methods to trivially understand the relationship, while directly only comparing to non-symbolic methods in a way that is not in some sense in the advantage of the non-symbolic methods.
 Additionally,  being more explicit about the potential weaknesses of the method, maybe empirically showing what happens with imperfect segmentation. The work is interesting, and I agree that this is a young field and the goal is *not* to produce state of the art results or outperform DRL methods. And is *not* to solve all the problems with symbolic methods at once, but to improve our understanding in this space. But I think the framing is not the right one in the current manuscript.